# A role for cortical interneurons as adversarial discriminators

**Ari S. Benjamin** *, **Konrad P. Kording**

Department of Bioengineering, University of Pennsylvania, Philadelphia, Pennsylvania, United States of America

* aarrii@seas.upenn.edu

## Abstract

The brain learns representations of sensory information from experience, but the algorithms by which it does so remain unknown. One popular theory formalizes representations as inferred factors in a generative model of sensory stimuli, meaning that learning must improve this generative model and inference procedure. This framework underlies many classic computational theories of sensory learning, such as Boltzmann machines, the Wake/Sleep algorithm, and a more recent proposal that the brain learns with an adversarial algorithm that compares waking and dreaming activity. However, in order for such theories to provide insights into the cellular mechanisms of sensory learning, they must be first linked to the cell types in the brain that mediate them. In this study, we examine whether a subtype of cortical interneurons might mediate sensory learning by serving as discriminators, a crucial component in an adversarial algorithm for representation learning. We describe how such interneurons would be characterized by a plasticity rule that switches from Hebbian plasticity during waking states to anti-Hebbian plasticity in dreaming states. Evaluating the computational advantages and disadvantages of this algorithm, we find that it excels at learning representations in networks with recurrent connections but scales poorly with network size. This limitation can be partially addressed if the network also oscillates between evoked activity and generative samples on faster timescales. Consequently, we propose that an adversarial algorithm with interneurons as discriminators is a plausible and testable strategy for sensory learning in biological systems.

## Author summary

After raw sensory data is received at the periphery, it is transformed by various neural pathways and delivered to the sensory cortex. There, neural activity forms an internal model of the state of the external world, which is updated appropriately by new information. A goal of learning, then, is to learn how to transform information into the appropriate representational form within the brain's internal model. Here, we look to artificial intelligence for new possible theories of how the brain might learn representations that resolve issues with previously proposed theories. We describe how one particular algorithm—adversarial learning—resolves a major issue with previous hypotheses relating to

**Data Availability Statement:** All code is available at https://github.com/KordingLab/adversarial-wake-sleep.

**Funding:** A.B and K.K. thank NIH EB028162 (www.nih.gov) for funding. The funders had no role in

study design, data collection and analysis, decision to publish, or preparation of the manuscript.

**Competing interests:** The authors have no competing interests to disclose.

recurrence. Furthermore, this algorithm resembles broad features of the organization and learning dynamics of the brain, such as wake and sleep cycles. Considering seriously how this algorithm would appear if implemented by the brain, we map its features to known physiology and make testable predictions for how neural circuits learn new representations of information.

## Introduction

The idea that the brain learns a generative model of the sensory world is now widespread in neuroscience and psychology. The ability to generate new patterns of activity similar to previous experience—the defining capability of a generative model—is evident, for example, in the spontaneous activity of dreaming sleep [1, 2] and in the spontaneous activity of waking states ([3], but see [4]). The concept of a generative model is also frequently invoked in the study of perception, as they can produce expectations about stimuli that are useful when stimulus information is noisy [5]. In motor control, internal models of sensory activity are also widely studied as a strategy for reducing motor errors [6, 7]. Together, these phenomena have provided converging support for the theory that brains build generative models of sensory activity.

Generative models are also key features within theoretical models of perception and sensory representations. In this framework, perception is posed as a process of inferring the properties of a hidden external world from noisy sensory data [5, 8]. This requires 'inverting' a generative model to perform Bayesian inference over its latent factors. Upon seeing a dark region on a two-dimensional screen, for example, one may infer the presence of a shadow of a three-dimensional object because this is most consistent with one's internal model of the world [9]. Many psychophysical and physiological experiments support this understanding of perception as inference over an internal generative model, and as a result this idea is now central to the modern theory of perception [10–16].

How are generative models learned? One popular theory holds that this is the purpose of dreaming sleep [17]. This is also the hypothesis of several classic computational models of sensory learning, such as the standard method for training Boltzmann machines [18, 19] and the Wake/Sleep algorithm for training Helmholtz machines [20]. In these theories, a generative model and approximate Bayesian inference over this model is learned by adjusting neural connections to align generative activity with typical stimulus-evoked activity. The goal is not necessarily to match single examples, but rather the overall probability distributions of activity in these two separate phases. Indeed, all algorithms for learning a generative model can be described as a process of matching the distribution of spontaneous activity with the distribution of stimulus-evoked activity.

In the last decade, a new family of algorithms has emerged that shares the strategy of aligning activity during two separate stages. Called Generative Adversarial Networks (GANs), this algorithm introduces a new neural network with the role of classifying, or discriminating, whether a given pattern of activity is real or self-generated [21]. A good discriminator acts as a teaching signal for the generative model, as the model can improve by opposing the discriminator. The generative model and discriminator thus learn 'adversarially'. In the machine learning literature, it has been noted that adversarial algorithms can also be used to learn approximate inference over a generative model [22–26]. In these papers, feedforward and feedback connections both learn by opposing a discriminator of whether pairs $(x, z)$ of inputs and hidden representations are driven top-down or bottom-up. If these two phases can be observed

separately, learning a discriminator is a powerful way to learn to perform Bayesian inference over a generative model.

Due to the suggestive separation of wake and dreaming-like phases, the question arises as to whether the brain might implement an adversarial algorithm. A key step in answering this question would be the identification of the discriminator in the brain. Thus far, the adversarial brain hypothesis has been explored by two papers which offer distinct possibilities. Gershman suggests that the phenomenology of subjective visual experience is consistent with the activity of a discriminator that could be implemented in prefrontal cortex [27]. Deperrois et al. propose that the feedforward connections in the sensory cortex itself act as a discriminator of generated activity in low-level areas [28]. Ultimately, detecting or confirming whether the brain learns adversarially requires identifying the discriminator and developing tests for such a possibility.

Here, we introduce the alternative possibility that discriminators are distributed within neocortical circuits as a dedicated cell type. Due to their local connectivity, such neurons would be classified as interneurons. The role of these cells, like all adversarial discriminators, is to provide a teaching signal that indicates how distributed activity in the surrounding population should change. The discriminator cells must themselves learn to be useful, and do so by learning to classify whether local activity is stimulus-evoked or self-generated.

This proposal is motivated by both computational and biological considerations. An adversarial algorithm can model a more flexible set of probability distributions relative to other ways in which the brain might learn approximate inference over a generative model. In particular, competing proposals cannot account for statistical dependencies between neurons, given a previous layer [20, 29, 30]. These algorithms instead require that neurons in one layer are conditionally independent given the previous layer. However, such dependencies are almost certainly present in the brain given the high degree of local recurrence. Adversarial algorithms avoid this problem as they do not require evaluating the likelihood of a network state under the generative model. The brain's heavily recurrent connectivity thus motivates the analysis of likelihood-free algorithms for generative modeling and inference and whether they are consistent with biology.

A central limitation of adversarial algorithms is their fragility. In practice, these algorithms do not scale well with the size of the input population, which may limit their biological plausibility. We document that this limitation can be partially removed when one considers a strictly layerwise architecture, and furthermore when activity is compared on a faster timescale of oscillations in addition to wake and sleep. This finding enables the use of discriminators that see only a local population, as do interneurons, and enables adversarial learning in brain-like architectures.

## Results

We first motivate the computational goal of inference over a generative model from the abstract goal of representation learning. We then derive a method to achieve these objectives with an adversarial algorithm and describe how this may be implemented in the brain. Finally, we evaluate the computational advantages and disadvantages of this learning strategy.

### Representation learning as distribution matching

This work adopts the view that the brain learns representations of the world in an unsupervised manner. In the standard formalization of representation learning, this requires learning two components. First, a generative model defines how representations relate to the inputs. This model should be optimized to ensure that representations convey as much information as

possible about the world [20]. Secondly, the brain should infer the representations that are consistent with new stimuli. Because sensory information is ambiguous, the brain should infer representations probabilistically. This involves performing Bayesian inference over the learned generative model. Further details of this standard formalization of sensory learning can be found in various textbooks [31, 32].

To take a simple example, one generative model might assume that data is composed of a mixture of discrete categories (Fig 1a), as in Gaussian Mixture Model. In this case one infers which category might explain new stimuli. Simultaneously, the generative model (e.g. the means and variances of each category) are optimized to match the data. Other examples of probabilistic representation learning differ in the form of their generative models, which span from linear transformations (e.g. in Probabilistic Principal Components Analysis) to the very complex hierarchical, nonlinear generative models that may be found in the brain.

It is crucial to note that because of the generality of this objective, the representations that are learned (i.e. the latent factors in the generative model) may represent nearly anything. Nothing in the objective itself ensures that representations will be 'useful' for any particular task. This further consideration of usefulness must instead be supplied by assumptions in the generative model (such as that the data falls into discrete categories). This generality is powerful in that it allows us to reason about representation learning in the brain without overly specifying what is learned. However, it also has the disadvantage that it is only partially determined.

For the purposes of explanation we will examine a bipartite system. Formally, we define a sensory system that consists of neural activity in a more abstract, 'higher' brain area, $\mathbf{z}$, and activity in a more concrete, 'input' brain area $\mathbf{x}$. The generative model maps $\mathbf{z}$ to $\mathbf{x}$, while the inference model maps $\mathbf{x}$ to $\mathbf{z}$.

A generative model defines a probability distribution over $\mathbf{x}$ for each $\mathbf{z}$. This distribution can be written as $p_\theta(\mathbf{x}|\mathbf{z})$, and it depends on the parameters of the top-down connections $\theta$. If one samples spontaneous activity from a prior distribution over representations $p_\theta(\mathbf{z})$, and propagates this spontaneous activity down to the input population, one obtains samples from

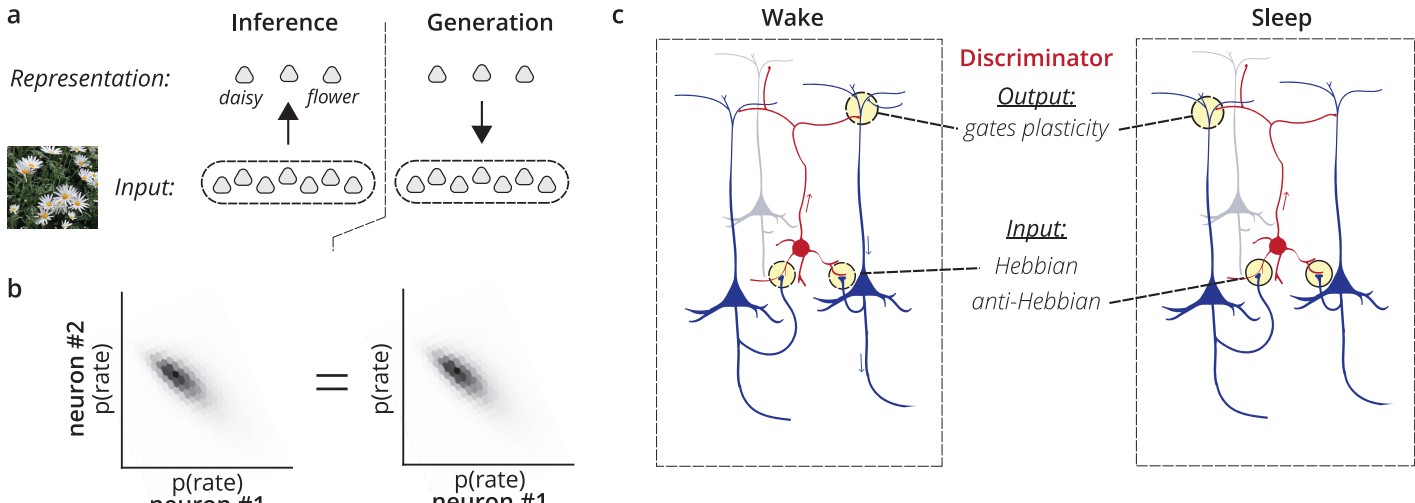

**Fig 1. Learning representations with a discriminating interneuron.** a) Representations are defined by a generative model that relates them to inputs. Organisms must learn to infer these representations from inputs (for example, to infer a visual category). b) This computational problem is equivalent to matching the joint probability distribution over inputs and representations during inference and generation. c) The brain may solve this problem with neurons that act as adversarial discriminators (red) of whether activity in neighboring neurons (blue) is dreamed or is stimulus-evoked. To achieve this, their input synapses would switch from a Hebbian to anti-Hebbian plasticity with the switch from wake to sleep.

a 'generative distribution' $p_\theta(\mathbf{x})$. This can be thought of as a 'dreamed' or 'fantasized' set of inputs. The objective of generative modeling is to match the generative distribution $p_\theta(\mathbf{x})$ with the empirical, stimulus-evoked distribution, such that generated and real samples cannot be distinguished. It can also be shown that this objective is equivalent to predicting inputs as best as possible, measured by the negative log likelihood ($-\log p_\theta(\mathbf{x})$), also known as the surprise [33]. Learning a generative model is thus equivalent to matching the distributions of generated and real activity in the inputs.

The objective of inference is also a distribution-matching problem. Given new stimuli, an organism may wish to infer what factors in its model of the world best explain sensory stimuli. This can be done with a bottom-up network, sometimes called an encoder or recognition model. Formally, the bottom-up network has the role of mapping an observed $\mathbf{x}$ to a distribution over $\mathbf{z}$, $q_\phi(\mathbf{z}|\mathbf{x})$. The objective of learning is to perform Bayesian inference over the generative model. Learning will adjust the parameters of this inference model, $\phi$, such that $q_\phi(\mathbf{z}|\mathbf{x})$ approximates the posterior distribution of $p_\theta(\mathbf{z}|\mathbf{x})$, i.e. the $\mathbf{z}$ could have generated an observed $\mathbf{x}$ under the generative model. If learning is successful, the bottom-up connections 'invert' the top-down connections and can infer self-consistent representations for new stimuli.

While inference and generation are two separate objectives of matching distributions, they may be combined into a single objective. We can write the *joint* distribution over $\mathbf{x}$ and $\mathbf{z}$ in the inference mode as $q_\phi(\mathbf{x}, \mathbf{z}) = q_\phi(\mathbf{z}|\mathbf{x})q(\mathbf{x})$. Here $q(\mathbf{x})$ denotes the probability that $\mathbf{x}$ occurs empirically; the use of $q$ reminds that it is a wake-phase distribution. Similarly, we can define the joint generative distribution, $p_\theta(\mathbf{x}, \mathbf{z}) = p_\theta(\mathbf{z}|\mathbf{x})p_\theta(\mathbf{x}) = p_\theta(\mathbf{x}|\mathbf{z})p_\theta(\mathbf{z})$. The joint distributions over $\mathbf{x}$ and $\mathbf{z}$ are aligned if and only if both the generative distribution is aligned ($p_\theta(\mathbf{x}) = q(\mathbf{x})$) and the inference distribution is aligned to the true posterior ($q_\phi(\mathbf{z}|\mathbf{x}) = p_\theta(\mathbf{z}|\mathbf{x})$). (If the joint distribution is aligned, so must any marginal distribution. The implication in the opposite direction can be proved by the definition of the joint distribution). Taken together, performing both inference and generation requires aligning $q_\phi(\mathbf{x}, \mathbf{z})$ with $p_\theta(\mathbf{x}, \mathbf{z})$. The joint probability distribution of pairs of *(real data, inferred representation)* must match the joint distribution of *(sampled representation, generated data)* pairs (Fig 1b).

## Aligning distributions with an adversarial algorithm

Any practical algorithm for sensory learning must maximize a measure of similarity between the inferred and generated joint distributions. The insight of the adversarial approach is that such a measure can be learned if one has a large number of samples of the two distributions. If no critic can discriminate the distribution from which a sample pulls, then the two distributions are the same. The goal is to change the distributions until they are indistinguishable in the eyes of the discriminator.

Adversarial algorithms represent an especially flexible strategy as they only require samples from distributions in order to align them. In contrast, many alternative algorithms for approximate inference (including Monte Carlo and variational inference methods) require the evaluation of the likelihood of a sample under either distribution. When the sample space is very high-dimensional and dimensions are correlated (as with neurons), this likelihood depends on all dimensions and may be quite complicated to compute. Being 'likelihood-free' thus eliminates this computational burden and makes adversarial algorithms attractive for approximate inference over complex generative processes [22, 23, 27].

Biologically, this makes adversarial algorithms compatible with the 'sampling hypothesis', which states that that neural activity represents a sample from the distribution encoding the brain's uncertainty over neural representations [34, 35]. In this scheme, greater uncertainty in inference corresponds to greater neural variability. When implemented in models of the

cortex, sampling procedures reproduce various neural phenomena such as task-dependent neuronal correlations, divisive normalization, stimulus-modulated noise variability, inhibition-dominated transients at stimulus onset, and oscillations [36, 37]. The strategy of representing uncertainty via sampling stands in contrast to neurons explicitly representing the parameters of probability distributions, such as their variance [38]. (It is worth noting that this is not a pure dichotomy, however, as one may also represent densities by sampling the component distributions of a mixture [39]). Committing to a sampling strategy to represent uncertainty, rather than an implicit distributional code, makes it more straightforward to implement an adversarial algorithm. The goal of representation learning becomes the alignment of the distributions of (stochastic) neural activity in the inference and generative modes.

The machine learning literature has extensively studied adversarial algorithms and their multiple potential formulations. The objective of the discriminator may be posed in multiple ways [40]. No one formulation is clearly more biologically relevant than others. For concreteness, and for the purposes of explanation and modeling, we consider the objective associated with the Wasserstein GAN [41]. However, the adversarial hypothesis is general to this implementation choice.

In the Wasserstein GAN formulation, the discriminator aims to maximize the difference of the average of the discriminator's activity in one phase with the average of its activity in the other. Representing the discriminator as $D(\mathbf{x}, \mathbf{z})$, its objective to maximize is:

$$\mathcal{L} = \mathbb{E}_{q_\phi(\mathbf{x},\mathbf{z})}[D(\mathbf{x},\mathbf{z})] - \mathbb{E}_{p_\theta(\mathbf{x},\mathbf{z})}[D(\mathbf{x},\mathbf{z})] \tag{1}$$

During the phase in which the network samples from the inference model $q_\phi(\mathbf{x}, \mathbf{z})$, the discriminator maximizes its output. During the phase in which the network samples from the generative model $p_\theta(\mathbf{x}, \mathbf{z})$, the discriminator minimizes its output. Note that Wasserstein GANs require an additional regularizing penalty so that the discriminator does not sharply vary with small changes in its input: $\nabla_{\mathbf{x},\mathbf{z}} D(\mathbf{x}, \mathbf{z}) \leq 1$. Pseudocode for this algorithm, also introduced in [22, 23] with minor differences, is shown in Algorithm 1. We discuss the biological predictions of this objective in the next section.

---

**Algorithm 1** Adversarial Wake/Sleep; see also BiGAN and ALI [22, 23]

---

1: **Initialize:** network parameters $\theta$ and $\phi$, gradient penalty $\lambda$, discriminator parameters $\theta_d$
2: **repeat**
3: **Wake phase (inference)**
4: $\mathbf{x} \leftarrow q(\mathbf{x})$ *Observe samples of inputs*
5: $\mathbf{z} \leftarrow q_\phi(\mathbf{z}|\mathbf{x})$ *Infer latent distribution and sample*
6: $\rho_q \leftarrow D(\mathbf{x}, \mathbf{z})$ *Discriminator observes network state*
7: $L_q \leftarrow \rho_q$ *Inference network loss*
8: $\alpha_q \leftarrow \lambda(\|\nabla_{\mathbf{x},\mathbf{z}} D(\mathbf{x}, \mathbf{z})\|_2 - 1)^2$ *Calculate gradient penalty*
9: $L_d \leftarrow -\rho_q + \alpha_q$ *Compute discriminator loss*
10: **Sleep phase (generation)**
11: $\mathbf{z} \leftarrow p_\theta(\mathbf{z})$ *Sample from the prior*
12: $\mathbf{x} \leftarrow p_\theta(\mathbf{x}|\mathbf{z})$ *Generate samples of fantasized inputs*
13: $\rho_p \leftarrow D(\mathbf{x}, \mathbf{z})$ *Discriminator observes network state*
14: $L_p \leftarrow -\rho_p$ *Generative network loss*
15: $L_d \leftarrow L_d + \rho_p + \lambda(\|\nabla_{\mathbf{x},\mathbf{z}} D(\mathbf{x}, \mathbf{z})\|_2 - 1)^2$ *Update discriminator loss*

16: $\theta_d \leftarrow \theta_d - \nabla_{\theta_d} L_d$ *Update discriminator parameters*
17: $\phi \leftarrow \phi - \nabla_\phi L_q$ *Update inference parameters*
18: $\theta \leftarrow \theta - \nabla_\theta L_p$ *Update generator parameters*
19: **until** convergence

---

**The problem of dimensionality and a solution with oscillations.** The adversarial wake/sleep algorithm aims to align probability distributions over all neurons in the system. This high dimensionality may prevent stable learning in very large networks. In the machine learning literature, applications of this algorithm to approximate inference have been limited to dimensions of **x** and **z** totalling tens of thousands of units at the largest [26]. This is orders of magnitude smaller than the number of neurons in sensory cortex, limiting this algorithm's biological plausibility.

One possible solution to this problem of dimensionality is to ensure that the global probability distribution can be factorized into several modules of conditionally independent populations. A common example of such a strategy is hierarchical architectures. In such networks, the activity in one layer depends only on the previous layer. As a result, the joint distribution over pairs of layers can be aligned without reference to the rest of the network.

An adversarial strategy for a hierarchical network could in principle employ many independent discriminators that see only pairs of layers. Even when matching pairs of layers, however, the input dimensionality is quite large. Discriminators must take as input the local population as well as the entire downstream (or upstream) population.

Here, to enable a further reduction of the dimensionality, we explore an additional approach that utilizes purely local discriminators that observe the local population in each layer of a hierarchical network. In a biological implementation, this would allow the role of the discriminator to be taken by interneurons (see next section). Interneurons are defined by their local connectivity, are ubiquitous in cortex, and have computational roles that are still uncertain. To enable the use of interneurons as discriminators, we again examine hierarchical networks but additionally introduce a phase of alternation whereby, in addition to wake and sleep, neurons compare their activity between phases of an oscillation. Note again that this strategy is applicable only when the network architecture is hierarchical and consists of many separate layers.

An algorithm in this setting of a hierarchy of layers can be derived by minimizing the KL divergence $D_{KL}(q_\phi(\mathbf{x}, \mathbf{z}) \| p_\theta(\mathbf{x}, \mathbf{z}))$. During inference, the activity in this hierarchy of layers can be written as,

$$q_\phi(\mathbf{x}, \ldots, \mathbf{z}_L) = \prod_{i=2}^{L} q_\phi(\mathbf{z}_i | \mathbf{z}_{i-1}) \, q_\phi(\mathbf{z}_1 | \mathbf{x}) \, q_\phi(\mathbf{x}).$$

Similarly, the generative distribution is,

$$p_\theta(\mathbf{x}, \ldots, \mathbf{z}_L) = p_\theta(\mathbf{x} | \mathbf{z}_1) \prod_{i=2}^{L} p_\theta(\mathbf{z}_{i-1} | \mathbf{z}_l) \, p_\theta(\mathbf{z}_L).$$

We assume that each layer has local recurrence and thus has a probability distribution that cannot be further decomposed into a product of distributions over single neurons.

The first step in our approach is to note that the the joint log-probabilities (which appear in the KL divergence) will take the form:

$$\log q_\phi(\mathbf{x}, \mathbf{z}) \quad = \sum_i \log q_\phi(\mathbf{z}_i | \mathbf{z}_{i-1}) \quad \bigg| \quad \log p_\theta(\mathbf{x}, \mathbf{z}) = \sum_i \log p_\theta(\mathbf{z}_i | \mathbf{z}_{i+1}) \qquad (2)$$

Because of this factorization in hierarchical architectures, the overall objective $D_{KL}$ will contain a term for each layer $i$. Each of these terms can be interpreted as the KL divergence

between the generative distribution and inference distribution over that layer, $\mathbf{z}_i$:

$$\underset{q_\phi(\mathbf{x},\mathbf{z})}{\mathbb{E}}\left[\log\frac{q_\phi(\mathbf{z}_i|\mathbf{z}_{i-1})}{p_\theta(\mathbf{z}_i|\mathbf{z}_{i+1})}\right] = \underset{\mathbf{z}_{i-1},\mathbf{z}_{i+1}\sim q_\phi(\mathbf{x},\mathbf{z})}{\mathbb{E}}\left[KL\big(q_\phi(\mathbf{z}_i|\mathbf{z}_{i-1})\parallel p_\theta(\mathbf{z}_i|\mathbf{z}_{i+1})\big)\right] \quad (3)$$

Thus, the global KL divergence in this architecture contains a series of local divergences over each layer $\mathbf{z}_i$ *in the inference phase*. The distribution of stimulus-evoked activity $q_\phi(\mathbf{z}_i|\mathbf{x})$ must be matched with the generative $p_\theta(\mathbf{z}_i|\mathbf{z}_{i+1})$ but, crucially, with $\mathbf{z}_{i+1} \sim q_\phi(\mathbf{z}_{i+1}|\mathbf{x})$ being a sample of inferred activity one layer up. This activity is then 'bounced back' to produce a generative sample $p_\theta(\mathbf{z}_i|\mathbf{z}_{i+1})$ immediately downstream. The layerwise KL divergence in Eq 3 aligns this 'bounced-back' distribution with the original forward distribution over $\mathbf{z}_i$.

In our case we will replace these layer-wise divergences with an adversarial objective. For each layer $i$, we can introduce a separate discriminator denoted by $D_i$. This approach thus allows the use of one discriminator that observes only the local population.

In the Wasserstein-GAN formalism introduced above, the objective would be:

$$\mathcal{L}_{O,\mathbf{z}_i} = \underset{q_\phi(\mathbf{z}_i|\mathbf{x})}{\mathbb{E}}\left[D_i(\mathbf{z}_i)\right] - \underset{q_\phi(\mathbf{z}_{i+1}|\mathbf{x})}{\mathbb{E}}\,\underset{p_\theta(\mathbf{z}_i|\mathbf{z}_{i+1})}{\mathbb{E}}\left[D_i(\mathbf{z}_i)\right] \quad (4)$$

The leftmost term represents a maximization of the expected activity of the *i*th discriminator during the inference phase. The right term represents the minimization of the expected activity of $D_i$ in a phase in which the neurons $\mathbf{z}_i$ are driven by feedback from one layer up. Thus the discriminator would observe alternating phases of somatic activity, with one phase corresponding to bottom-up inputs and one phase corresponding the top-down prediction. This algorithm is described step-by-step in Algorithm 2.

---

**Algorithm 2** Adversarial oscillation algorithm

---

1: **Initialize:** Hierarchical network parameters $\theta$ and $\phi$, gradient penalty $\lambda$, discriminator parameters $\theta_d$
2: **repeat**
3: $\mathbf{z}_0 \leftarrow q(\mathbf{x})$ *Observe samples of inputs*
4: **for** $i = 1$ to $L$ **do**
5: $\mathbf{z}_i \leftarrow q_\phi(\mathbf{z}_i|\mathbf{z}_{i-1})$ *Infer samples given previous layer*
6: $\rho_i^{(q)} \leftarrow D_i(\mathbf{z}_i)$ *Layerwise discriminator observes encoded samples*
7: $L_i^{(q)} \leftarrow \rho_i^{(q)}$ *Inference network loss*
8: $\alpha_i^{(q)} \leftarrow \lambda(\|\nabla_{\mathbf{z}_i} D_i(\mathbf{z}_i)\|_2 - 1)^2$ *Calculate gradient penalty*
9: $L_i^{(d)} \leftarrow -\rho_i^{(q)} + \alpha_i^{(q)}$ *Compute discriminator loss*
10: $\hat{\mathbf{z}}_{i-i} \leftarrow p_\theta(\mathbf{z}_{i-1}|\mathbf{z}_i)$ *Generate one layer down*
11: $\rho_{i-1}^{(p)} \leftarrow D_{i-1}(\hat{\mathbf{z}}_{i-1})$ *Discriminator of $i-1$ observes generative sample*
12: $L_{i-1}^{(d)} \leftarrow L_{i-1}^{(d)} + \rho_{i-1}^{(p)} + \lambda(\|\nabla_{\hat{\mathbf{z}}_{i-1}} D_i(\hat{\mathbf{z}}_{i-1})\|_2 - 1)^2$ *Update discriminator loss at $i-1$*
13: $L_{i-1}^{(p)} \leftarrow -\rho_{i-1}^{(p)}$ *Generative layer loss*
14: **end for**
15: $\mathbf{z}_L \leftarrow p_\theta(\mathbf{z}_L)$ *Handle top layer; sample from the prior*
16: $\rho_L^{(p)} \leftarrow D_L(\mathbf{z}_L)$ *Discriminator of $L$ observes sample of prior*
17: $L_L^{(d)} \leftarrow L_L^{(d)} + \rho_L^{(p)} + \lambda(\|\nabla_{\mathbf{z}_L} D_L(\mathbf{z}_L)\|_2 - 1)^2$ *Update discriminator loss at $L$*

18: $\theta_d \leftarrow \theta_d - \nabla_{\theta_d} \sum_{i=0}^{L} L_i^{(d)}$ *Update discriminator parameters*
19: $\phi \leftarrow \phi - \nabla_{\phi} \sum_{i=1}^{L} L_i^{(q)}$ *Update inference parameters*
20: $\theta \leftarrow \theta - \nabla_{\theta} \sum_{i=0}^{L-1} L_i^{(p)}$ *Update generator parameters*
21: **until** convergence

---

Several algorithms which bear a close similarity to this have appeared in the machine learning literature. In the application of adversarial inference to hierarchical networks, the closest parallel work is Hierarchical Adversarially Learned Inference (HALI) [42]. There are two main differences of this oscillatory algorithm with respect to HALI. First, HALI employs a single discriminator that observes the entire hierarchy of latent vectors, rather than the series of layer-wise discriminators introduced here. Second, HALI only employs the generative model in a purely offline phase, whereas the oscillatory algorithm generates samples one layer down during the inference phase. HALI is thus best thought of as the application of the Adversarial Wake/Sleep algorithm to a hierarchical network, rather than an oscillatory algorithm that leverages the structure to factorize the discriminator. The oscillatory algorithm we have described here is essentially a hierarchical version of the VAE-GAN [43]. There are also similarities of this algorithm with the Stacked GAN as studied by Huang et al. [44], although this works with a pretrained inference model (the encoder) while here the inference model is learned adversarially. Despite these differences, these papers provide important proofs of the effectiveness of this general strategy in artificial systems.

In the Discussion we evaluate the plausibility of this interpretation of oscillations in light of known physiology.

## Interneurons as discriminators of inference and generative phases

The specific hypothesis of this paper is that specific cell types in the cortex act as discriminators of activity to guide sensory learning (Fig 1). These cells receive connections from local populations of cells, and then project back to these local populations to control plasticity. As the local population shifts between internally- and externally-biased modes of information, these cells learn to discriminate, i.e. to have higher firing rates in one phase compared to the other. In the section we describe this circuit and the key identifying attributes of the discriminating cells.

**Plasticity of synapses targeting discriminator cells.** An identifying property of the hypothesized discriminator cell type is that its learning rule should change sign with wake and sleep. Specifically, the connections onto a discriminator cell should change to produce a high firing rate on waking activity and low firing rate on sleeping activity (or vice versa). This is a simple restatement of the objective in Eq 1. As a consequence of this rule, a cell's firing rate effectively becomes an estimate of whether local activity is more characteristically sleep-like or wake-like.

With only a small assumption about the nonlinearity the input/output function of the discriminator cell, it can be seen that the connections onto a discriminator cell exhibit plasticity that is Hebbian in one phase and anti-Hebbian in another. Specifically, this is the case when discriminator cells approximately take a linear-nonlinear form such that $D(\mathbf{x}, \mathbf{z}) = \sigma(W_x \mathbf{x} + W_z \mathbf{z})$ for a monotonic nonlinearity $\sigma$. This linear combination followed a nonlinearity is a standard, if oversimplified, model of the firing rate of many neuronal responses. It has the property that the gradient with respect to the synaptic strengths $W$ is a Hebbian multiplication of pre- and postsynaptic information, e.g.:

$$\nabla_{W_x} D(\mathbf{x}, \mathbf{z}) = \mathbf{x} \odot \sigma'(W_x \mathbf{x} + W_z \mathbf{z}). \tag{5}$$

The adversarial rule thus requires adjusting the weights with the product of these terms in one phase (a Hebbian rule), while adjusting them with the negative of the product (an anti-Hebbian rule) in the other phase.

An adversarial learning rule may also specify additional attributes of synaptic plasticity. In Eq 1, we chose the objective of a Wasserstein GAN, which has the additional requirement that the discriminator varies smoothly with its input, also known as a Lipschitz constraint. This is

defined formally as $\nabla_{\mathbf{x,z}} D(\mathbf{x}, \mathbf{z}) \leq 1$. This can be seen as a penalty (or constraint) upon synaptic weights and the nonlinearity of the discriminator. For example, if the discriminator is a linear-nonlinear neuron, the Lipschitz constraint can be rewritten as $W_x \odot \sigma'(W_x\mathbf{x} + W_z\mathbf{z}) \leq 1$, which specifies that the neuron must have a small synaptic strength ($W_x$) and the neuronal nonlinearity must have a small derivative for any input. Biological neurons might achieve these properties via multiple strategies. Natural physical constraints upon the size of a synapse, as well as the achievable receptor density, represent an upper bound on synaptic strength that, if plasticity is otherwise unaffected, would resemble a simple weight clipping procedure. Surveys of synaptic volumes reconstructed via electron microscopy indeed find an upper limit of synaptic size [45]. Alternatively, this constraint could present as a penalty (or regularization term) upon synaptic size, akin to the gradient penalty of [46]. The effect of this penalty would be an inverse dependence of induced LTP upon the initial synapse strength. Indeed, classic plasticity induction experiments in cortical and hippocampus slices report such a dependence in excitatory cells [47–50], demonstrating its plausibility in a potential discriminatory cell. The Lipschitz constraint required in a Wasserstein GAN thus naturally aligns with biological constraints and penalties upon synaptic size and efficacy.

It is important to recognize that the Wasserstein GAN, together with its Lipschitz constraint, is only one of many adversarial objectives that the brain may implement with discriminating cells. It is worth examining what features of learning are general to the adversarial hypothesis versus what are specific to the Wasserstein formulation. For example, we may alternatively hypothesize that cells optimize the 'standard' adversarial objective first proposed by Goodfellow et al. [21], and applied to approximate inference by [22, 23]:

$$\mathcal{L} = \mathop{\mathbb{E}}_{q_\phi(\mathbf{x,z})} \left[ \log D(\mathbf{x}, \mathbf{z}) \right] + \mathop{\mathbb{E}}_{p_\theta(\mathbf{x,z})} \left[ \log \left( 1 - D(\mathbf{x}, \mathbf{z}) \right) \right].$$

This objective bears a broad similarity to that of the Wasserstein GAN but with key differences. For example, the required plasticity is not symmetric with phase. Deriving a plasticity rule via evaluating the derivative with respect to $W_x$ yields,

$$\nabla_{W_x} \mathcal{L} = \mathop{\mathbb{E}}_{q_\phi(\mathbf{x,z})} \left[ \mathbf{x} \odot \frac{\sigma'(W_x\mathbf{x} + W_z\mathbf{z})}{\sigma(W_x\mathbf{x} + W_z\mathbf{z})} \right] + \mathop{\mathbb{E}}_{p_\theta(\mathbf{x,z})} \left[ \mathbf{x} \odot \frac{-\sigma'(W_x\mathbf{x} + W_z\mathbf{z})}{1 - \sigma(W_x\mathbf{x} + W_z\mathbf{z})} \right].$$

This update remains Hebbian and anti-Hebbian for the respective phases. In the inference phase (left term) the update is an expectation over a product between the presynaptic $\mathbf{x}$ and a postsynaptic term. This is a Hebbian expression. In the generative phase, the update remains a negative of a product between a presynaptic and a postsynaptic term, although it is no longer an exact negative reflection of the inference phase's learning rule. Thus, different adversarial objectives present similarly at a higher level of abstraction and share the broad feature of switching from Hebbian to anti-Hebbian learning for the inputs to simple discriminating cells.

**Gating of network plasticity by discriminator cells.** The computational purpose of the discriminator cell is to control plasticity in the inference and generative networks such that they produce the same distribution of activity over neurons. According to the adversarial theory, this requires these networks *minimize* the same objective $\mathcal{L}$ that the discriminator aims to maximize. Synapses in the generative model or the inference model should thus change based on their downstream effect upon the discriminator. This entails a credit assignment problem.

There are multiple ways that the inference and generative networks could solve this credit assignment problem in a biologically plausible manner. What is common to all strategies is that the activity of the discriminator somehow gates or affects plasticity in these feedforward or feedback synapses. One possibility, for example, is for synapses to follow the REINFORCE

rule [51]. In the case of the inference connections, the gradient of the Wasserstein objective (Eq 1) is,

$$\nabla_\phi \mathcal{L} = \mathbb{E}_{q_\phi(\mathbf{x},\mathbf{z})} \left[ D(\mathbf{x}, \mathbf{z}) \nabla_\phi \log q_\phi(\mathbf{x}, \mathbf{z}) \right]. \tag{6}$$

This strategy prescribes a reward-modulated learning rule, with the discriminator's activity replacing reward. Implementing this rule would require that discriminator cells project back to cells in the local population and multiplicatively gate a predictive plasticity in their input synapses.

Research into biologically-plausible credit assignment strategies is an active field with multiple competing hypotheses [52]. Because of this multiplicity, we wish to separate our adversarial hypothesis from this issue and do not commit to any one model. However, it is worth describing a few possibilities. For example, the brain might encourage a 'weight symmetry', in that the axonal synapses from the discriminator affect the surrounding cells with a strength that reflects the synaptic strengths from those same cells to the discriminator. A predictive, Hebbian rule is one strategy that encourages this symmetry [53, 54], and a symmetry may naturally emerge when these feedback synapses are fixed [55]. Other categories of strategies that do not require such symmetry include spike-based causal inference [56] or neuromodulatory approaches [57]. The cells that input to the discriminator cells may use any one of these methods to ensure that their plasticity is in proportion to their effect upon the discriminator.

Regardless of how the credit assignment problem is solved, the effect of the discriminator upon plasticity is similar. As the brain transitions from wake to sleep, the gating effect of the discriminator upon local plasticity must also change sign from negative to positive. This parallels the switch from Hebbian to anti-Hebbian plasticity in the synapses onto the discriminator. This phasic sign inversion of plasticity is a strong prediction of an adversarial algorithm and is common to all choices of objectives and credit assignment strategies.

**Summary.** Above we have described how the brain may solve the computational goal of generative modeling and approximate inference with a cell type that discriminates between activity in wake and sleep. This cell may alternatively (or additionally) discriminate between phases of an oscillation of whether neurons are driven bottom-up or top-down. If the spike rate of this discriminating cell is a simple (linear-nonlinear) function of its inputs, this cell will have Hebbian plasticity in one phase and anti-Hebbian in the other. The activity of this cell type must also gate local plasticity in proportion to the effect of those cells upon the discriminator. This gating effect will likewise switch from a negative gain to a positive gain with the phase of sensory processing. In the Discussion we survey the literature to suggest potential candidates for this discriminating cell, as well as experiments that could identify this cell more conclusively.

## Computational analysis

The adversarial framework has already been shown to be capable of approximate inference over a generative model in neural network settings [22–25, 42, 43]. These publications provide a general proof of principle of the effectiveness of adversarial approximate inference.

In this section we present additional neural network simulations to demonstrate the capabilities of an adversarial approach, including both advantages and disadvantages. It is important to note that artificial neuronal network models differ from biology in many aspects. Our approach is not to include all possible biological details, or to claim that each computational component in these models has a direct corresponding element in the brain. For example, these 'neurons' do not spike and have no dendritic trees, ion channels, or cell types. The

architectures are not directly modeled on brain connectivity. We assume that biologically plausible credit assignment is solved and use backpropagation to calculate gradients. Finally, we make several arbitrary choices about the batch size, learning rate, and optimizer that are not meant as hypotheses about biology. In these and in other aspects our models should not be interpreted as earnest simulations of the brain. Rather, each experiment proves a general capability (or failure) of the algorithm which together can help adjudicate whether it is powerful enough for sensory learning in the brain.

**Approximate inference over recurrent generative models.** One crucial test of biological feasibility that has not yet been verified in the machine learning literature is whether this algorithm can work perform approximate inference over recurrent generative models. This capability is important to demonstrate as it would mark a significant advantage over competing theories of how the brain performs approximate inference.

Many previous theories of how the brain might learn generative models and approximate inference require calculating likelihoods. For example, many variational inference algorithms proposed as brain models that minimize the 'surprise' for new data, which is the negative log likelihood of that data under the generative model [13, 19, 29, 30]. However, local recurrence complicates this likelihood-based strategy because it introduces dependencies between neurons. To circumvent this problem, these previous theoretical models prevent such dependencies by eliminating the problematic recurrence, in effect sacrificing biological realism for computational tractability.

In Fig 2, we present a trivial task designed as the minimal case in which local recurrence causes dependencies that complicate variational models. We define a target population (cells B and C), which also driven by external stimulation from a 'teacher' cell T. Additionally, we

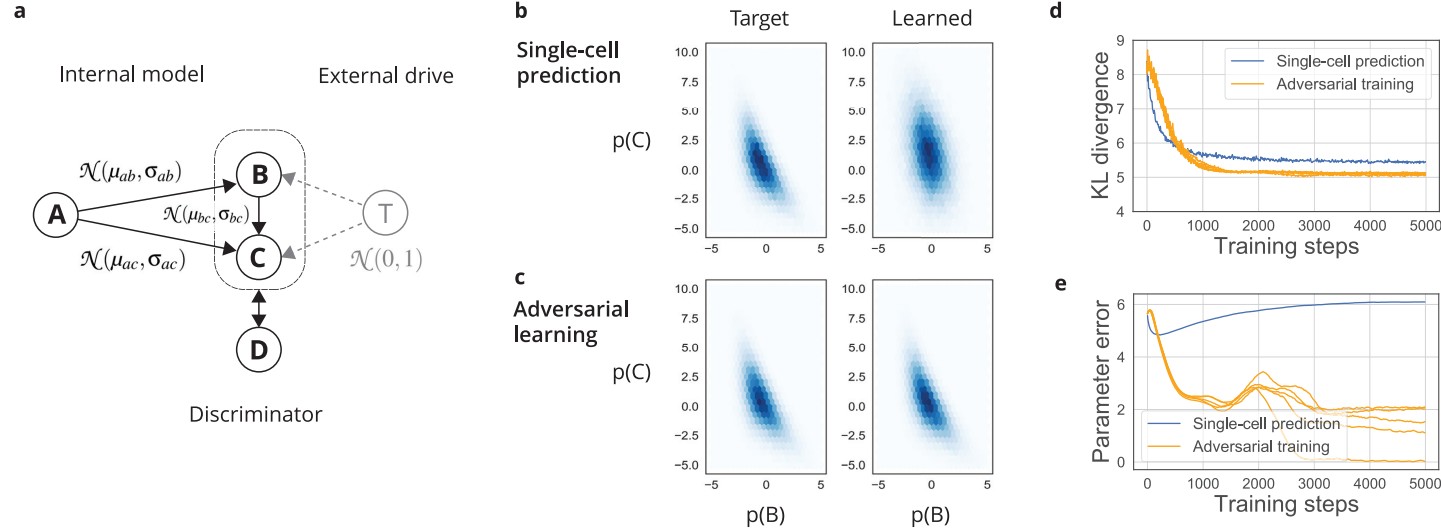

**Fig 2. A simple case illustrating the problem of predicting activity in a population with internal connections.** a) We created a toy task in which the goal is to adjust the connections away from neuron $A$ to better model the activity of neurons $B$ and $C$. The "target population" $[B, C]$ is also driven by an external teaching signal from neuron $T$. All connections evoke Gaussian-distributed postsynaptic responses, and all neurons sum their inputs linearly. b) When learning the $A \rightarrow B$ and $A \rightarrow C$ connections by predicting the postsynaptic neuron (e.g. maximizing log $p(B|A)$, ignoring recurrence), the predictions fail to match the joint distribution. The left plot shows the true joint distribution $p(B, C)$, which is correlated and non-Gaussian due to the $B \rightarrow C$ connection internal to the target population, while the right plot shows $\mathbb{E}_A[p(B, C|A)]$. c) The adversarial strategy successfully aligns the distribution. The discriminator sees $B$ and $C$ and gates plasticity at the predictive connections from $A$. d) The alignment can be quantified with the KL divergence of the binned histogram between the learned and target distribution. We plot the trajectories from 5 random runs; all networks share the same initialization. e) Another measure of success is the distance of the parameters of the outward connections from $A$ from their optimal value, which we call the parameter error.

define a neuron (A) whose output is used to model the distribution of B and C induced by T. If learning succeeds, the joint distribution $p(B, C)$ should be the same when driven by T as when driven by A. However, neurons B and C interconnect. This causes dependencies in their distribution given neuron A (Fig 2b).

In a variational solution to this problem, the synapses from A would learn by attempting to maximize the log probability of observing $B$ and $C$ given $A$: $\log p(B, C|A)$. If B and C were independent given A, then the $A \rightarrow B$ synapses and the $A \rightarrow C$ synapses could learn with local information (a Hebbian rule). However, recurrence between $B$ and $C$ introduces a dependency, meaning that $\log p(B, C|A) \neq \log p(B|A) + \log p(C|A)$. Thus $B$ and $C$ cannot be modeled independently. Synapses from A need access to all $A$, $B$, and $C$'s activity in order to learn. Indeed, because of the dependency between B and C given A, learning a model of B and C by predicting them separately causes poor learning (Fig 2b). Ultimately, the problem posed by recurrence for variational methods in the brain is that synapses now also require nonlocal information about many other neurons' activity.

To solve this problem adversarially, one can introduce a discriminator. This discriminator sees the activity of A, B, and C, and controls plasticity at the synapses from A onto B and C. Learning proceeds by alternating between phases. In the 'external' phase, B and C are driven by A, whereas in the 'internal' phase they are driven by the hidden cell T. The discriminator learns to classify these phases by attempting to increase its activity in one phase and decrease it in another. The synapses from A attempt to affect the activity of the discriminator in the opposite way. This adversarial strategy allows the synapses from the source neuron A to correctly model their distribution (Fig 2c). Overall, this occurs because the discriminator observes the entire network state and summarizes it in a low-dimensional way that remains useful for learning.

We next aimed to demonstrate this algorithm in a higher-dimensional neural network with recurrence. We created a neural network with both a recognition model and generative model of raw inputs (Fig 3), both of which are stochastic, nonlinear, and recurrent. The generative model and inference model are identical in architecture and are as follows. First, a 2-layer fully-connected neural networks output the mean and variance of a diagonal Gaussian over the latent $\mathbf{z}$ or inputs $\mathbf{x}$, which is then sampled. Then, this sample is given to a stochastic network of the same architecture, the output of which is interpolated with the original sample via an arithmetic mean. This step models a single step of recurrence. As a result of the nonlinearity and interpolation, the conditional distributions $q_\phi(\mathbf{z}|\mathbf{x})$ and $p_\theta(\mathbf{x}|\mathbf{z})$ are not Gaussian. This recurrence induces dependencies among neurons in each layer, given the previous layer.

We trained this network with the adversarial wake/sleep algorithm. In the offline 'sleep' phase, random representations are sampled and the network generates novel inputs $\mathbf{x}$ from its learned model. In the 'wake' phase, the network sees a batch of digits and the network produces representations $\mathbf{z}$ by sampling from $q_\phi(\mathbf{z}|\mathbf{x})$. The goal of learning is to produce the $\mathbf{z}$ that could have generated those inputs, and to be able to generate realistic inputs from the prior. To assist in this goal, a discriminator sees both inputs and representations and aims to classifies pairs of $(\mathbf{x}, \mathbf{z})$ as being from wake or sleep.

By training against this discriminator, we found the network learned to generate realistic digits from its prior (Fig 3d). This demonstrates that adversarial approach is sufficient to train a model of the inputs $p_\theta(\mathbf{x})$ even in recurrent architectures. However, the empirical evaluation surfaced an issue with the adversarial wake/sleep algorithm. While generation is of good quality, reconstructing digits produces different digits than those input (Fig 3e). This suggests that inference is not optimal. Reconstruction maps to the manifold of MNIST digits, but elsewhere. This is despite the optimal solution being perfect inversion. A wide search over learning rates, weight decay, and the $\beta$ parameters of the Adam optimizer did not produce better reconstructions.

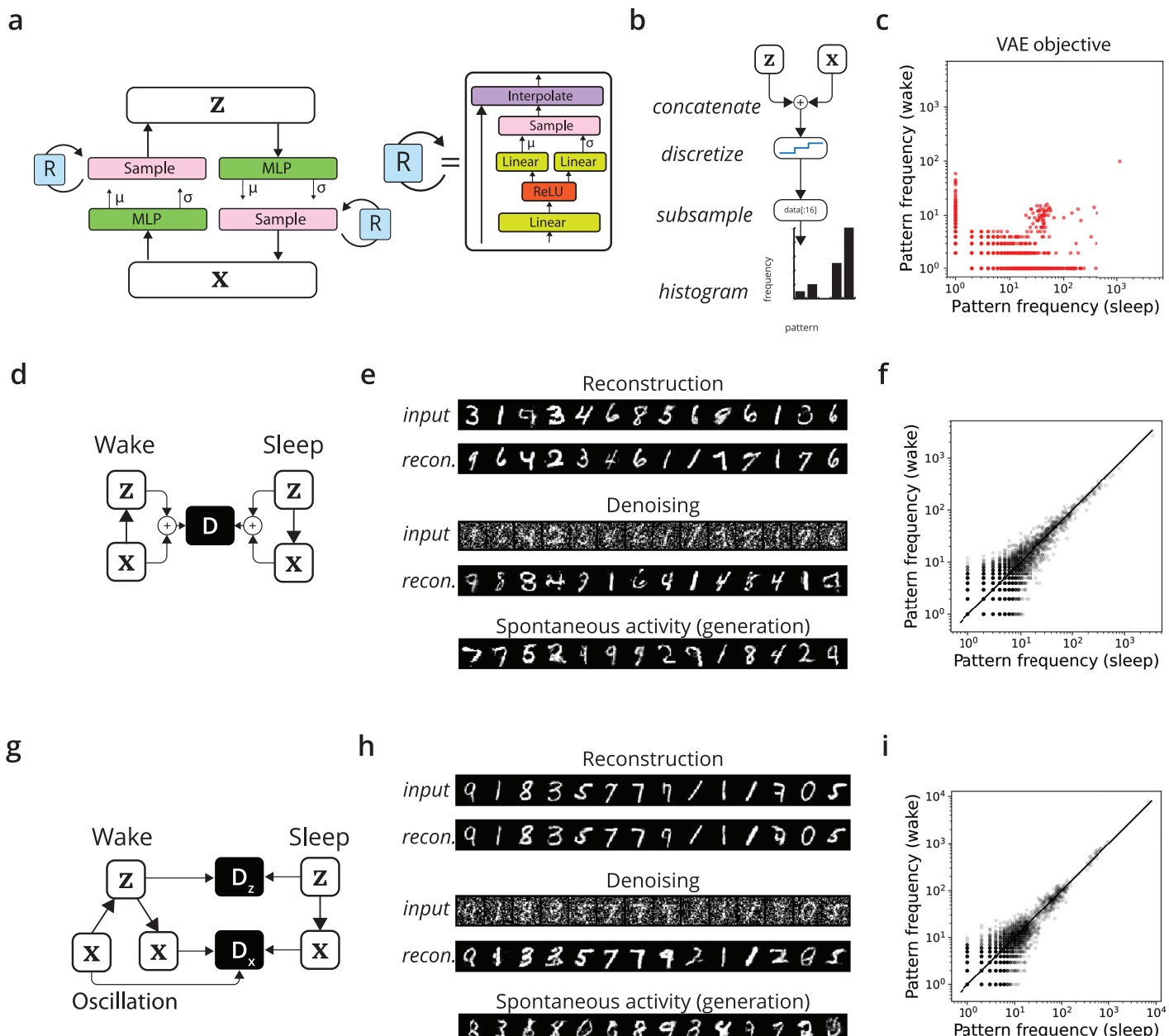

**Fig 3. Approximate inference in a generative model of MNIST digits using a recurrent, stochastic autoencoder.** a) In both inference and generation, samples are processed through a multilayer network that parameterizes a multivariate Gaussian. The network is then sampled and undergoes nonlinear stochastic recurrence. b) Performance metric: Following Berkes et al. [3], we empirically verify if the joint distributions of 'sleep' and 'wake' phases match by comparing the frequency over a binned histogram. We quantize the vectors of **x** and **z** by rounding to the nearest integer and quantify the histograms of the frequency of each pattern across 32 units' activity when observing/generating MNIST digits. Perfect performance corresponds to all points lying on the $y = x$ diagonal. c) As a baseline, we train the architecture with the standard variational autoencoder objective function (a.k.a. the ELBO), approximating the inferred distribution as a Gaussian and essentially ignoring the stochastic recurrence. This approach unsurprisingly fails, with patterns in 'sleep' occurring at different frequencies than in 'wake'. d) In the adversarial wake/sleep algorithm, a global discriminator observes both **z** and **x** during wake and sleep phases. e) This algorithm enables realistic generation but often results in poor reconstruction. f) The frequency of patterns over a subsample of units is much more similar in sleep and wake than in (c). g) The oscillatory algorithm: During the wake phase, inputs are reconstructed through the generative model, and the discriminator attempts to classify real from reconstructed. The discriminator also aims to classify wake from sleep. h) Reconstruction improves due to oscillations without negatively affecting generation. i) Matched patterns for the oscillatory algorithm.

Unfortunately, the same factors that make this task interesting (its dimensionality and recurrence) also make it difficult to provide a quantitative evaluation of quality. Direct measures like the log-likelihood or KL divergence are not tractable, and kernel density estimation is prohibited by the dimensionality. However, a quantitative measure is possible if the data is downsampled. We thus employed the approach of Berkes et al. [3] and empirically quantified the overlap of the wake and sleep joint distributions. After quantizing the inputs and latents to the nearest integer and subsampling to 32 units (16 $\mathbf{x}$ and 16 $\mathbf{z}$), we counted how many times each unique pattern over the 32 units appeared in wake or sleep. The quality can be compared by visual inspection, as the closeness of patterns to the diagonal.

For comparison, we also trained this same recurrent architecture with a strategy that ignores recurrence and attempts to estimate the likelihood of the inferred representation under the generative prior. Specifically, we trained this architecture using the standard variational autoencoder (VAE) loss function, also known as the variational free energy or ELBO [58]. This is possible because the probability distributions in inference (absent recurrence) are Gaussian and can be reparameterized. However, this strategy can only work by ignoring how recurrence changes the inferred probability distribution to something non-Gaussian. Thus, the loss function relies on an incorrect assumption. Unsurprisingly, we found that this significantly hampered training. The histograms of 32 random units appear severely different; patterns that occur during inference rarely occur during sampling (Fig 3c). This underscores the importance of accounting for recurrence and its consequences for the form of conditional probability distributions.

We next trained the network with the oscillatory objective. There are two differences with this approach. First, the discriminator is local to each layer, a feature that is more biologically plausible (Fig 3g). Second, in the processing of information, an input is transformed into a representation $\mathbf{z}$, and then that representation is fed into the internal model to produce a reconstructed input $\mathbf{x}'$. The local discriminator attempts to classify the original input $\mathbf{x}$ from $\mathbf{x}'$. In addition, we also employed the approach of the VAE-GAN of using the hidden layers of the discriminator as a metric between inputs and reconstructions [43]. We found that this approach was much more stable (Fig 3h).

This experiment demonstrates the effectiveness of adversarial representation learning even in settings where recurrence complicates stochastic representations. Our intention is not to establish a machine learning baseline; very similar algorithms exist in the literature (e.g. [22, 43]). Rather, it is to convey the computational advantage of this algorithm for approximate inference in a more similar algorithmic setting that might be found in the brain.

## Discussion

Here we have described how discriminating interneurons might mediate the learning of sensory representations in the brain. This cell-type specific computational role can be derived from the perspective that the brain learns probabilistic representations of its inputs. Since representation learning is equivalent to distribution-matching (i.e. the probability distributions of neural activity should be identical whether stimulus-evoked or self-generated), this can be achieved with an adversarial algorithm utilizing a discriminator. This hypothesis provides a concrete implementation of the idea that the brain learns internal models of the world via switching between externally- and internally- driven modes of processing, both on the timescales of wake and sleep as well as on the timescale of oscillations [17].

The plausibility of this adversarial hypothesis is informed in part by computational arguments. Unlike variational inference algorithms, adversarial algorithms are 'likelihood-free' in that they do not require any calculation of the likelihood of a network state under the

generative model. This is advantageous, as such likelihoods are extremely complicated to obtain in recurrent architectures. The choice of a likelihood-free algorithm frees theoretical analysis from the restricted and recurrence-free architectures of typical variational inference models, such as the bipartite graph [13, 19, 20, 29, 30]. This flexibility with respect to network architecture makes adversarial algorithms appealing as biological hypotheses due to the ubiquitous recurrence found in the brain.

However, adversarial algorithms also bring computational difficulties. As is widely appreciated (or bemoaned) by practitioners, adversarial algorithms are extremely fragile. Small changes to seemingly innocuous parameter settings like the learning rate lead to large changes in performance. Underlying this fragility is the game-theoretic nature of adversarial learning. Success depends on a careful balance between the discriminator and the generator. Without an effective theory capable of maintaining this balance, machine learning practice has settled through trial and error upon a narrow set of parameters that work well. This fragility has several consequences for the evaluation of biological hypotheses. First, it makes it difficult to simulate more biologically-realistic models, such as spiking neural networks, as this setting diverges from the narrow region of architectures and optimizers proven to be successful by the machine learning literature. Furthermore it prevents the effective falsification of theories through simulation as it is never clear whether a failure might be fixed by small, seemingly arbitrary parameter choices. Finally, this fragility also makes it somewhat less plausible that evolution has found effective settings, though there possibly exist stabilizing factors. For example, it is interesting that divisive normalization, which is a ubiquitous form of local recurrence in the cortex [59], is also known to stabilize GAN training [60]. At the very least, because fragility worsens with increasing dimensionality, it is unlikely that the brain adversarially aligns the distribution over all neurons' activity. For this reason our hypothesis focuses on local discriminators that align the distribution of neural populations containing perhaps only a few tens of thousands of neurons, akin to a cortical column.

It is useful to contrast our algorithm with related hypotheses in which the brain learns adversarially [27, 28]. Gershman similarly proposes that adversarial learning may underlie probabilistic inference and generative modeling in sensory cortex, which is the objective considered here and is identical to the ALI and BiGAN approach [22, 23]. Our biological proposal differs, however, in that we introduce cell types as discriminators whereas Gershman suggests that prefrontal cortex may play this role. We also consider the effect of adversarial fragility with dimension and as a solution propose purely local discriminators that leverage cortical oscillations. Deperrois et al. propose that it is the feedforward path of sensory cortex which plays the role of a discriminator of low-level inputs. Furthermore, in only aligning the activity distributions of the input areas, this proposal differs with ours and Gershman's, which aim to perform probabilistic inference over higher areas and thus align activity in all of sensory cortex. Deperrois et al. also propose a three-phase learning algorithm of wake, REM sleep, and NREM sleep, which serve to calculate an additional reconstruction error over inputs (wake phase) and a reconstruction error over top-level latents that are replayed by the hippocampus and then perturbed (in the NREM sleep phase). Our algorithm does not include these reconstruction errors or hippocampal replay. Overall, these central differences serve to highlight the importance of local discriminators with small input populations as well as preserving the high-level computational goal of probabilistic inference over generative models.

## Biological evidence and candidate cell types

The adversarial brain hypothesis presents several strong predictions about connectivity, learning, and sensory processing. How plausible is this concept? Is there enough biological evidence

to suggest a specific cell type as a discriminator? In the following sections, we review the relevant literature and outline criteria to refine this search. Ultimately, this hypothesis can only be confirmed through experiments testing its specific predictions (refer to Fig 4).

Due to their local connectivity, these neurons would be classified as non-projecting neurons or interneurons by definition. Although non-projecting excitatory neurons exist, the majority of interneurons are inhibitory. An additional identifying factor is their ability to control local plasticity, particularly during the critical period. Somatostatin-positive interneurons meet this criterion [61], as do $5\text{-}HT_{3A}R^+$ cells in Layer 1 [62]. To influence plasticity in this manner, the discriminating interneurons must have specific control over signals that regulate plasticity in those cells. For instance, spike bursts elicit significant calcium releases (thereby affecting plasticity) but are strongly regulated by the action of apical-dendrite targeting inhibition [63]. In this way, dendrite-specific inhibition can gate plasticity within cortical networks in a manner consistent with the effect of discriminating cells.

A further prediction of this theory is that the effect of discriminating cells on local plasticity must change from positive to negative gain with the overall phase of sensory processing (e.g., wake to sleep, or phases of an oscillation). For this to happen, there must be a signal that communicates the phase of processing and affects plasticity accordingly. Acetylcholine, a neuromodulator implicated in the control of sleep and initiation of dreaming, is a strong candidate for this role [64–66]. Interestingly, acetylcholine also plays a crucial role in cortical plasticity [67, 68]. For example, if acetylcholine is prevented from being released [69], cortical learning is impaired. Tellingly, this also occurs if acetylcholine's effect upon somatostatin-positive interneurons is blocked [61], suggesting that it is a key modulator of these interneurons' affect on

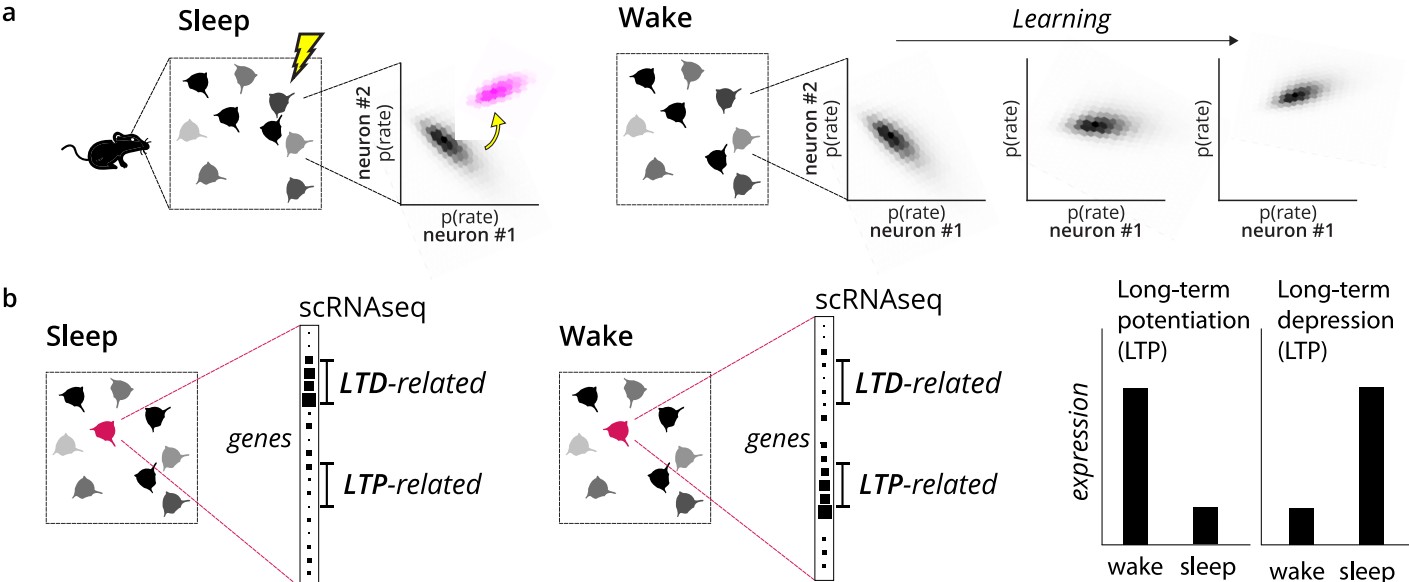

**Fig 4. An experimental program to test the adversarial hypothesis.** a) A key first step is to test whether the brain's learning algorithms actually do align the distribution of activity in wake and sleep. This would begin with identifying neurons in chronic recordings that have the same distribution of activity patterns in wake and sleep. A subset of these neurons could then be perturbed during sleep to produce a new distribution (pink). Over the course of several nights, it should be the the case that the distribution of stimulus-evoked activity should align with the perturbed distribution. This experiment could identically be run by perturbing the waking activity and observing learning in sleep. b) A next step is to identify if this alignment is mediated by discriminating interneurons. This might begin with a survey of gene transcription in interneurons in wake and sleep, with the prediction that some class should exclusively transcribe genes associated with postsynaptic long-term depression (LTD) in sleep, and genes associated with postsynaptic long-term potentiation (LTP) in waking states. Blocking this synaptic plasticity in only this cell type should prevent the alignment observed in the experiment in (a), even though it concerns different cell types.

plasticity. This suggests a biologically plausible strategy whereby acetylcholine, or another neuromodulator, might mediate the switching of the discriminator's affect upon local plasticity with the phase of processing.

Connections to local discriminatory interneurons should exhibit a plasticity rule that switches polarity with the phase of processing, from a simple Hebbian to an anti-Hebbian rule. While there is no known cell-type specific evidence for this phenomenon, there is evidence for such switches in plasticity at more general scale. Synaptic strengths across the cortex are known to homeostatically increase during wakefulness and decrease during sleep [70–72]. Phasic changes in plasticity have also been observed at the timescale of oscillations, at least in hippocampus [73–75]. These phenomena are broadly consistent with the adversarial hypothesis's prediction of plasticity rules that change synchronously with the phase.

It is useful to compare the properties of an adversarial algorithm with others in the theoretical literature that also use phasic mode switches for learning internal models [13, 30, 76]. In particular, there is a a notable difference in prediction for sub-cellular and dendritic responses. Here, in the feedback phase of the oscillation, top-down feedback should drive somatic activity. Other algorithms for Bayesian inference require that feedback only be integrated in dendritic compartments for comparison with bottom-up activity, with the difference between the two driving learning [13, 30, 76]. In reality, feedback into apical dendrites can have a large effect on somatic activity [63]. The adversarial framework can thus explain why oscillations affect somatic activities rather than just apical dendrites.

A fundamental requirement of this hypothesis is that the brain operates in two distinct phases of information processing: one for stimulus-evoked inference and another for self-generated activity. A strong candidate for the second phase is dreaming sleep, and particularly REM sleep. Numerous computational studies have suggested that cortical activity during sleep represents samples from a generative model [17, 28, 66, 77]. However, this perspective remains controversial. Much of the evidence is phenomenological; dreams exhibit many properties of the world, such as meaningful objects and sounds, and are rarely mere repetitions of past experiences [78]. There is also evidence of a similarity between neural activity in dreams and waking states. For instance, it is possible to decode visual imagery in dreams from neural activity using algorithms trained on waking activity [2]. Intriguingly, a recent study found that neural ensembles in V1 are replayed in sleep, although with greatest similarity in slow-wave sleep [79]. However, some argue that the surreal and temporally discontinuous nature of dreams indicates they are purposefully perturbed in some manner, and thus not straightforward samples from the brain's generative model [80]. This aspect of dreams must be reconciled with the generative model framework. If the brain only aligns local distributions of activity between wake and sleep, for example, this might allow for unusual pairings of activity at the level of distant brain areas. The answer to this broader question about the statistical nature of activity in dreaming sleep is directly relevant to the plausibility of the adversarial hypothesis.

Alternatively, instead of dreaming sleep (or in addition to), we have proposed here that oscillations could represent the necessary mode switches between bottom-up and top-down signals. For a literature review of this possibility, we direct readers to the perspective offered in Honey et al. [17]. One example line of evidence is that the visual cortex is more sensitive to external visual stimuli at the troughs of individual cycles of alpha oscillations [81, 82]. Another consequence is that unexpected stimuli should evoke stronger observable oscillations in activity than expected stimuli. After all, for expected (i.e., well-modeled) stimuli, the top-down predictions are accurate enough to be indistinguishable from bottom-up signals, and no observable change is expected with a phase shift. Therefore, we anticipate a correlation between the power of the oscillation and surprise. In the auditory cortex, MEG and EEG studies show increases in gamma power in response to unexpected auditory stimuli

[83, 84], omissions of expected musical beats [85], and unexpected mismatches between auditory and visual cues [86]. In the hippocampus, both theta and gamma ranges increase after unexpected stimuli [87]. A final prediction is that the oscillations should ascend up sensory hierarchies. As information feeds forward it appears as a traveling wave. Indeed, oscillations in the gamma range have been found to ascend up the visual hierarchy [88]. These numerous findings together support the notion that oscillatory waves could indeed represent switches between generative and inference modes that could be leveraged for learning.

## Possible experiments

The confirmation of this hypothesis would require a dedicated research program, but it is not beyond the capabilities of current experimental methods. A series of experiments that confirm progressively more specific predictions may make it possible to positively identify a discriminating interneuron and thus firmly establish whether the brain learns with an adversarial strategy. Here, we outline one such research program.

The first experiment in this series is the most general. It must first be demonstrated that the brain aligns the probability distribution between neural activity in a phase of sleep (or oscillation) and stimulus-evoked activity. A further study investigating this could closely resemble that of Berkes et al. [3], which compared the distribution of activity in the ferret visual cortex when eyes were open and closed. Instead of comparing spontaneous closed-eye activity, one could compare activity during phases of LFP oscillations or during sleep. These distributions should match in adult animals. This is perhaps the simplest experiment and is likely achievable with current recordings, as it only requires a set of neurons to be recorded longitudinally over multiple hours, or even days, while simultaneously tracking the phase of processing (oscillation or wake/sleep). A positive outcome in this experiment would provide key evidence for the broad theoretical framework, tracing back to at least the Wake/Sleep algorithm [20], that the brain learns inference over a generative model by switching between phases of processing.

Perturbations would enable a deeper test of this general framework (Fig 4a). During sensory learning, one could selectively silence neural activity during a stage of sleep or half of an oscillatory cycle. This perturbs the generative distribution. By tracking neurons over multiple days, one could then observe if activity changes in the waking or opposing phase to match that perturbed distribution. If this capability is confirmed, it would indicate that the probability distribution of activity in these two phases is actively regulated to match. This key feature of the framework suggests that the brain learns representations (i.e., probabilistic inference over factors in a generative model) by comparing two phases. This general capability would then provide the starting point for experiments into the cell-type specific mechanisms that lead to it.

The identity of the discriminator is the major element that differs between our proposal and that of Gershman and Deperrois et al. [27, 28]. To identify a discriminating interneuron among the many non-projecting cell types, it may be most efficient to search for signatures of the plasticity switch with phase. One technique for this search is genetic methods for observing single-cell gene transcription, as these are now relatively inexpensive and can be scaled to many neurons [89]. Using single-cell RNA sequencing (scRNAseq) taken at various phases of sleep, for example, one could list candidate cell types whose expression of factors relating to postsynaptic LTP and LTD is dominated by the sleep cycle. This unique prediction may be observable in datasets available with today's technology [90]. This would provide a first-line survey for discriminating cells (Fig 4b). A further step is to establish the causal role of these candidate cell types upon local sensory plasticity. By optogenetically perturbing their activity while tracking plasticity, one should see that their gating effect on plasticity shifts with phase.

Similar experiments could be done for the discriminators proposed by Gershman and Deperrois et al., namely, the excitatory projection cells in the sensory cortex (in the case of Deperrois et al.) or in the prefrontal cortex (in the case of Gershman). Through such perturbations, one could effectively determine the identity of the discriminator.

**Conclusion.** This study investigated the possibility that a subtype of cortical interneurons might serve as discriminators in an adversarial algorithm for representation learning in the sensory cortex. By examining the characteristics and plasticity rules of these interneurons, we provide a foundation for experiments that might confirm this theory and help reveal the brain's algorithm for sensory learning. Our findings suggest that adversarial learning offers potential advantages over previous theories, particularly in terms of modeling correlated statistics and addressing issues related to recurrence. However, we also identify limitations with network size scalability, which can be partially mitigated by incorporating faster timescales and oscillations between evoked activity and generative samples. This theoretical work paves the way for future research and experiments that could further our understanding of the cellular mechanisms underlying the learning of sensory representations.

## Methods

### Software and hardware

The experiments presented were coded in Pytorch v1.8. All experiments were run on NVIDIA GeForce GTX 1080Ti GPUs. Running on a single card, the experiments take about 5 minutes to train the MNIST architecture. Searches over training details required around 100 times these totals.

### MNIST architecture

For the MNIST task [91], we used the train set for training and the test set to display reconstructions. MNIST is licensed CC BY-SA 3.0. Before training and generation, we rescaled inputs to the range $[-1, 1]$.

The architecture of both the inference and generation networks was a fully-connected, two-layer neural network with 512 hidden units and 128 output units (or 784*2 in the case of generation). The hidden nonlinearities were Exponential Linear Units (ELUs). Half of the output units specify the standard deviation and half the mean of a diagonal Gaussian over the outputs. This iswasthen sampled stochastically, and this sample fed to nonlinear recurrence with the same architecture as the inference/generation networks. Finally, the output was recombined via an arithmetic mean with the inputs.

The prior distribution over the latent variables was a standard normal distribution. Sampling new inputs involves sampling from this prior, followed by processing through the generation network.

### Discriminator architecture

We trained two adversarial algorithms in the MNIST architecture: one purely wake/sleep and the other additionally with an oscillation.

In the wake/sleep algorithm, the discriminator was an ensemble of discriminator cells, each of which sees a random subselection of the entire network state. Each of these 'cells' is itself a 3-layer neural network. The network had linear outputs, and each internal layer had 512 hidden units and a nonlinearity of a LeakyReLU unit with negative slope 0.2. The final discrimination decision was an average over all cells in the network state.

In the oscillatory algorithm, the discriminator was identical except that the connectivity of the discriminator cells was restricted such that each cell only observes a selection of units from either the inputs **x** or the representation **z**.

## Algorithmic details

For all adversarial objectives we used the Wasserstein-GAN [41] with a gradient penalty of $\lambda = 1$ [46]. As is standard practice, the gradient penalty is applied to a random interpolation of the inputs and the generated samples.

The high-level algorithm in the wake/sleep MNIST task is unchanged from previous applications of adversarial representation learning [22, 23]. In the 'wake' phase, inputs were fed through the inference network to obtain samples of inferred representations. These were paired and given as input to the discriminator. The objective function of the discriminator in this phase was the mean of its output across the batch, whereas the objective function of the inference network was the negative of the discriminator's mean. In the 'sleep' phase, the prior over representations was sampled and inputs are generated through the generative network. These were paired and given as input to the discriminator. The objective function of the discriminator in this phase was the negative of the mean of its output across the batch, whereas the objective function of the inference network was the discriminator's mean. In both phases the discriminator was additionally regularized with a Lipschitz gradient penalty.

The oscillatory algorithm in this MNIST architecture was similar to the VAE-GAN algorithm [43]. In the oscillatory algorithm, the 'wake' phase differs in that an input is transformed into a representation **z**, and then that representation is fed into the internal model to produce a reconstructed input **x′**. The local discriminator attempted to classify the original input **x** from **x′**. In addition, as in the VAE-GAN, the hidden layers of the discriminator were used as a metric between inputs and reconstructions. The 'wake'-phase objective of the inference and generative networks was to oppose the discriminator and also minimize the distance between the input and the reconstructions under the discriminator's metric. This special 'wake' phase objective was added in sum to the wake/sleep objective described above.

For comparison, we also trained the MNIST architecture with a standard ELBO. This objective function minimizes the reconstruction error, while simultaneously minimizing the KL divergence of the inferred representation from the generative prior. This requires an estimation of the entropy of the inferred distribution, which is not available analytically because of the stochastic recurrence. As stated in the Results, we ignore this recurrence and take the entropy as that of the Gaussian parameterized before recurrence.

## Optimization

For all tasks we used the Adam optimizer [92] with $\beta_1 = 0.5$, $\beta_2 = 0.99$, and weight decay of $2 \times 10^{-5}$. The batch size was 512 except for the Moving MNIST task, which for memory reasons was 32. The learning rate of the inference and generation networks was $10^{-4}$ and the discriminator's learning rate of $4 \times 10^{-4}$. These learning rates decreased by a factor of 0.96 between epochs 200 and 250. The total number of training epochs for all tasks was 400. These hyperparameters were chosen from previous literature (specifically [22]). Random hyperparameter searches were also conducted, but these did not overly improve results. In general we found that adversarial training is very fragile to the balance of architecture, learning rate, and optimizer between the generator and discriminator. Architectures were tuned by hand by visually examining the quality of reconstructions on the training set and generations from random noise.

### 4-neuron toy experiment

In this experiment, all nodes are modeled as linear-summing units with stochastic Gaussian outputs. The initial weights are chosen at random from a standard normal distribution. To enable error propagation through the Gaussian stochasticity, we employed the reparameterization trick.

For the non-adversarial baseline, we aimed to minimize the log probability of the two nodes (when driven by T) under the connections from A, assuming their independence. We trained the weights of A using gradient descent with a learning rate of 0.001.

For the adversarial algorithm, we created a discriminator, consisting of a small neural network with a single hidden layer of 20 units and a Leaky ReLU nonlinearity with a negative slope of 0.2. We employed the Wasserstein GAN objective function with a gradient penalty of 0.1 and optimized all weights using RMSProp with a learning rate of 0.001. When comparing this algorithm to the non-adversarial baseline, we started with the same random initialization of weights. The remaining stochasticity arises from the sample generation process.

To assess the performance, we report the empirical KL divergence between the binned hexagonal histograms of the 2-dimensional distribution over B and C. We used a grid size of 25 bins per side, with an extent from [-5, 5] on the x-axis and [-5, 10] on the y-axis.

## Author Contributions

**Conceptualization:** Ari S. Benjamin.

**Formal analysis:** Ari S. Benjamin.

**Funding acquisition:** Konrad P. Kording.

**Investigation:** Ari S. Benjamin.

**Methodology:** Ari S. Benjamin.

**Project administration:** Konrad P. Kording.

**Software:** Ari S. Benjamin.

**Supervision:** Konrad P. Kording.

**Visualization:** Ari S. Benjamin.

**Writing – original draft:** Ari S. Benjamin.

**Writing – review & editing:** Ari S. Benjamin, Konrad P. Kording.

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
