## [Decision Letter · Decision Letter 0]

21 Dec 2022

Dear Benjamin,

Thank you very much for submitting your manuscript "An algorithm for learning to infer in recurrent networks" for consideration at PLOS Computational Biology.

As with all papers reviewed by the journal, your manuscript was reviewed by members of the editorial board and by several independent reviewers. In light of the reviews (below this email), we would like to invite the resubmission of a significantly-revised version that takes into account the reviewers' comments. A main theme from these reviews is that the biological constraints need to be considered more carefully in order to make a persuasive case that this is something the brain might be doing.

We cannot make any decision about publication until we have seen the revised manuscript and your response to the reviewers' comments. Your revised manuscript is also likely to be sent to reviewers for further evaluation.

Sincerely,

Samuel J. Gershman

Section Editor

PLOS Computational Biology

Reviewer's Responses to Questions

**Comments to the Authors:**

Reviewer #1: Summary

=======

The authors consider the problem of learning a generative model of data in a variational inference setting.

In contrast to other works, which often assume conditional independence of neurons in each layer of the generator, they consider networks in which each layer can be recurrent.

They argue that since biological networks are recurrent, this is a more natural setting for (generative) models of brain computation.

To solve the computational issue of optimizing a generative model in the presence of recurrence, and to make learning more compatible with biophysical constraints, they hypothesize that additional "discriminator" neurons in each layer together with alternating bottom-up and top-down drive on the generative neurons implement an adversarial learning mechanism.

They demonstrate with a toy tasks, and also using moderately challenging machine learning datasets that their suggested principle can successfully learn generative models in various network architectures also in the presence of recurrence.

The manuscript is technically sound, the authors clearly understand the mathematical and computational principles underlying (amortized) variational inference and adversarial learning.

The presentation is moderately clear, but should be improved (see detailed comments); most importantly the methods should be significantly extended (or and appendix should be added) with derivations and details on the formal setup of the different tasks (e.g., the graphical models).

My main concern is that I could not clearly understand what novel insights the authors would like to communicate (this issue is potentially already highlighted by the, in my opinion, rather vague title). Is this a machine learning paper that has important implications for biology? Is this a biology paper heavily inspired by machine learning? If the main contribution is in terms of algorithms (large parts of the introduction and results read like it is), then I believe paper is severely lacking a discussion of the existing literature and comparison to existing models; a brief Google scholar search reveals works from 2016 which already consider adversarial learning with recurrent networks. If the main contribution is on the biology side, then I would expect significantly more discussion of the challenges of implementing the model in biological substrates (to justify the authors claim "This extension of the Wake/Sleep algorithm offers a bridge between the computational objective of Bayesian inference and the biological implementation in synaptic plasticity."). Specifically, from a biological perspective I see the following issues:

- What would a "learning rule" (which the authors mention) look like according to this framework and how would it be implemented in a biological substrate?

- requiring a precise, distributed timing (oscillation algorithm)

- relying on backprop (through time!)

- using LSTMs

- requiring careful (hyperparamter) tuning to overcome instabilities of adversarial learning

- using divisive normalization

- using ADAM

- using convolutions

The simulations the authors provide are all rather abstract (wrt biology) and seem to assume that all of these complications are somehow solved in the biological substrate.

From a methodological perspective, evaluating the quality of the learned generative model is mainly left to the reader via visual inspection; I think the manuscript would benefit from more quantitative evaluation measures.

If I may be so bold to make a suggestion: it seems to me that the main contribution of this manuscript lies in the extension of wake-sleep to non-Gaussian, non-factorizing distributions using adversarial learning. If novel (please check!!!), this is an important contribution to the statistical learning literature. In this case, I can envision a clear, compelling manuscript focused on this topic which could be published at appropriate high-impact (ML) venues. In an additional (perhaps "perspective"?) manuscript the authors could then highlight the implications of this new machine learning model for learning in biological substrates.

Please receive additional notes for the authors below.

General notes

- motivation rests on prediction in time, but model does prediction "in space"; please keep the two separate; how can you motivate the latter?

- why are learning a generative model and inference in it meaningful objectives for the brain to solve? first, a generative model does _not_ allow prediction of _future_ inputs (in particular not the model considered here); second, why transform sensory input into a "high-level" representation; all computation implies non-increasing information, so what would the brain try to achieve?

- much of "Comparing phases of an oscillation" (ll278-319) is not really results and would maybe better suited for the discussion

- how does the discriminator gate plasticity biologically?

Line notes

- ll35-36: or they just compare new experience with old experience? or they just learn not to be surprised? a lot of (implicit) assumptions in the first sentence! if not explained further at least add references

- l38: /if/ the previous hypotheses are correct, then learning adapts the predictive model

- l40: what kind of benefits?

- l42: "...predict must..."? merge issue?

- ll42-46: kind of complicated way to explain (Bayesian) inference; simplify?

- l57: "abstract"/"concrete" in what sense?

- l69: what is an internal model? a generative network? a predictive network? it would be useful to clearly distinguish the two (the former transforms high-level representations to low-level activities, the latter predicts future high-level representations from past high-level representations)

- ll76-79: motivation is quite weak ("recurrent connections exist, hence correlations expected"); first, there _are_ definitely correlations (see any paper on neural correlations) and second, one could argue from a functional perspective: models that capture additional correlations are more powerful; can you give an example where this additional computational power would be useful?

- l82: "incoming synapses" from where? top-down? recurrent? would be helpful to be explicit here

- ll87-95: could you briefly summarize the "justification" you allude to? could you provide an intuition for why such adversarial learning could be useful?

- l93: who is "they"?

- l101: "...share the By contrast..."; merge issue?

- l109: "consequences"

- l110: "their similarity to known results"; sounds a bit weak, maybe formulate stronger if you are talking about experimental results? for example "explain empirical findings"?

- ll124-125: using q for both the true data distribution and the approximate posterior is quite confusing; is it ever possible to match the true distribution in a finite-data scenario?

- l134: how is q(x,z) defined? is it the combination of the approximate posterior with the true data distribution as a prior?

- l134: explain a bit more why matching the joint distributions fulfill the two objectives, i find it non-obvious (e.g., it is a necessary or sufficient condition? how to arrive from matching joint distributions at the two original objectives?)

- l143: i don't understand: also VI matches distributions, so why use the joints here?

- l152: what is a transport problem? so far you have only stated that p(activity neuron i) != p(activity neuron i|activity of all [other] neurons); also, maybe don't assume everyone is familiar with the weight transport problem

- ll155-165: if possible (according to journal), some math would really help to get the point across, and maybe a figure

- l175: "one solution" ;)

- l182: define "easily"; how are LSTMs implemented in the brain? how is backprop through time solved?

- l186: it should probably read something like "two distribution are better aligned if a discriminator can not tell samples from them apart"

- l215: and Deperrois et al.

- l221ff: you mention learning rules, but only show objectives; what would the learning rules look like? what quantities are used in the learning rules and how do they get there, i.e., are they local? who teaches the discriminator? what is the architecture of the discriminator? does learning in your model need backprop (if so, how do you solve all the issues associated with bio bp?)? do you need the regularizer? if so how is it implemented in bio?

- l243: how correlated are these increases/decreases in synaptic strength with the source neuron type? i guess your model would predict opposite changes for projections from exc/inh neurons? is there evidence for this?

- l249: i'm a bit concerned about the timescale here: if oscillations have periods of ~10ms, and the neurons response time is also on the order of 1-10ms, doesn't the somatic output "mix up" information from the two phases? to me it seems like you would need longer timescales if you want to separate the two phases and want to take the biophysical dynamics of neurons into account

- l255: maybe mention here explicitly that this a difference to the "standard" approach; in your case, the joint distribution of a layer _can not_ (trivially) be further decomposed into a product of distributions over neurons and that you can thereby capture arbitrary (conditional) statistical dependencies between neurons in one layer!

- l264ff: an important difference to wake sleep is the fact that there's is a discriminator, right? also not sure what "the above setting" and "this algorithm" refer to

- l270 (eq 5): could you provide a derviation of this objective from eq 2; it doesn't seem completely obvious over which distributions the expectations should be taken; for example, didn't you assume a perfect inference model and shouldn't the first expectation of the second term be over the join q(z{i+1}, x)?

- l280ff: very cool interpretation of oscillations!

- l285ff: it seems to me that two things are mixed up here; first, oscillations switch non-discriminator neurons from top-down to bottom-up drive and provide a teaching signal to the discriminator(?); and second the activity of the discriminator itself is interpreted as the oscillation; the oscillation can hardly be teaching the discriminator and be generated by it simultaneously, can it? i think this needs to be explained better

- l308f: which plasticity switches sign? that on "inference/generative network" neurons or those on discriminator neurons?

- l321: which algorithms? it's not clear that the two previous sections describe two different algorithms; rather to me it seems like wake-sleep + discriminator describes an abstract ML model and the oscillation part describes a potential biologically plausible implementation

- l324: which objectives? (see previous comment)

- l334ff: i'm quite confused: i would've expected that learning adjust connections in the internal model (T->A,B) rather than in the external drive (A->B,C)!? here it seems like the internal model doesn't learn anything but rather the representation is fixed and the stimulus is somehow mapped to this representation; the logic behind the setup should be explained a bit more i think; in addition, aren't the results obvious: if you assume independence where there isn't, you learn worse?

- l362: what does "during recurrence mean"? do you run the network until it reaches a steady state? for Xms? one "step"?

- l375ff: adversarial learning in recurrent architectures was done before in ML, maybe mention this; do they also find that it doesn't work very well for inference?

- l383: "biology may have found solutions to stabilize this approach" is a bold claim and would benefit from support (futher arguments why this should be the case or references)

-l393: "signifying that the inference network maps to the support of the generative posterior" seems like a bold statement given these results; your objective now explicitly contains a reconstruction loss, so it is not surprising that reconstructions become better; still latent representations of data could reside in a different part of the latent space than what you use for spontaneous generation; more analysis needed if you want to make a strong statement here; also, would be good to quantify the generation/reconstruction quality, maybe FID?

- l400: even with Berkes' approach you rely on visual inspection of the plots; could you distill it into a single number that could be compared between algorithms?

- l408: how would divisive normalization be implemented? with an additional population? how about error transport, locality of information etc?

- l414: wasn't this explored in the ML literature before? is it surprising that it works/what is the novelty here? how bioplausible is this setup?

- l419: interesting point about delays but first, this should come way earlier, and second this should be discussed more

- l429: non-Gaussian or non-diagonal or both?

- l430: how bioplausible is any of this?

- l448: "It is general [with regard to]"?

- l457: is this objective really easier? in hierarchical networks gradient descent on the variational free energy doesn't seem much more complicated than your proposal; if you want to make this point, explain explicitly why yours is easier; indeed just thinking about the specific timing requirements of switching discriminator targets in a continuous-time system like across cortex seems daunting

- l467: in Deperrois et al., plasticity in wake is also training the encoder (in addition to the discriminator)

- l477: what does "classify oscillations" mean?

- l483: is it necessary that the plasticity switches sign? couldn't you also just switch the target? i assume the theory suggests the latter; if so, are the two equivalent?

- ll490-498: this is a rather general prediction from learning/using generative models during sleep, right? it doesn't seem specific to your hypothesis

- ll499-504: cool experiment! seems extremely challenging, and I'm not sure it's obvious why this would be the case; maybe expand a bit (encoder tries to invert generator)?

- ll505-512: not sure i understand; maybe chat with an experimentalists to see whether its clear what you are suggesting?

- ll514-517: yes! maybe this point could already be picked up earlier in the manuscript; i feel its essential

- l526ff: i think this should also be mentioned earlier; i was wondering in the results quite a bit how you optimize, in particular because you mention "plasticity rules"

- l532: maybe end the manuscript on a stronger note?

- l564: explain your implementation of recurrence more

- l592: not mandatory, but i prefer all details in the manuscript; if everything is in one place it's easier to look it up and one doesn't need to assume the code is always available

Figure notes

- figure 1: maybe keep all recurrent connection of the network in "our hypothesis" in addition to discriminator connections; not sure what "target neurons" are; do you just mean "neurons"?

- figure 2: is b the same as the first panel in c? maybe only keep one? in e, why is the adversarial error initially so small? should both errors start at the same point?

- figure 3: in a, not sure what the "combine" operation is, please explain; in b,c please explain the three different tasks (reconstruction, denoising, spont. acivity); in c, why are there three arrows into the discriminator, what is discriminated against what? possible to indicate in figure?

- figure 4: why reconstruct from z1? isn't that rather easy? how about reconstructions from the deeper layers? are these good reconstructions? are these good generated images? provide at least some baseline to compare to (plain autoencoder? wake/sleep?) and some quantitative evaluation of the images (FID?)

- figure 5: c, explain the different tasks, and again, are these results good or bad?

Reviewer #2: In this manuscript, Benjamin and Körding propose a Wasserstein GAN model for inference in recurrent cortical circuits. Their model proposes alternating "wake" and "sleep" phases, in which the discriminator is alternately driven by bottom-up inputs and top-down predictions. They then argue that this algorithm bears some similarity to cortical learning phenomena, and provide proof-of-principle simulations to demonstrate its efficacy. I find the broad idea of the proposed algorithm somewhat interesting. However, I think the results of the present manuscript fall short of what is promised---namely, "the missing link between a widely hypothesized computational goal of learning and its implementation in neural systems"---and I do not believe it is suitable for publication in its present form.

Below, I give more detailed comments, roughly separated by importance but not ordered within each category. I would like to preface my remarks with the caveat that my knowledge the literature on sleep or on cortical oscillations is superficial at best, so I cannot offer useful comments on those aspects of the paper.

Major comments:

1. The algorithmic challenge the authors aim to address is the complicated dependency structure induced by recurrent connectivity. However, I found the presentation of their proposed solution hard to follow on first reading. I would suggest that the authors separate the mathematical construction of their model from the discussion of its possible biological implementation. Concretely, I would propose the following organization: (i.) introduction of the problem of recurrence and of the proposed WGAN algorithm at an abstract, mathematical level; (ii). proof-of-concept simulations to demonstrate that the proposal is an effective solution; (iii). mapping onto biology, existing evidence, and proposed experimental tests. Moreover, to allow the reader to understand precisely what is being proposed, I suggest that the authors provide pseudocode for their algorithms (as in, for example, the original WGAN paper by Arjovsky, Chintala, and Bottou). This could be included in a box. The concrete problem addressed by the paper should also be stated precisely in the abstract, rather than with vague generalities. I believe this reorganization would improve the readability of the paper, and make it easier to evaluate its contributions.

2. The authors mention the fact that the discriminator in (2) requires access to the full network state, which is not biologically plausible. They do not discuss in detail the fact that the algorithm in (3-5) shares similar limitations: depending on the size of the clusters in (3), each local discriminator must receive input from a potentially very large population. Therefore, the efficacy of their approach would seem to rely on each layer of neurons being relatively small. For their algorithm to be a compelling proposal, the authors must demonstrate that this is not an unreasonable requirement. Namely, how large might one expect the clusters of neurons implied by Figure 1 to be? Though doing so is beyond the scope of the present manuscript, another requirement would be to show that biologically-constrained architectures can successfully be trained on adversarial tasks, given the known instabilities in GAN training (as discussed, for instance, in the paper by Donahue and Simonyan that the authors themselves cite). This is likely a particularly strong limitation if the layers are large. With this in mind, I think it is also important for the authors to discuss the plausibility of stable biologically-plausible WGAN training in greater depth than the cursory mention in Lines 526-532.

3. Based on my reading of the paragraph starting on Line 236, it seems the authors propose that the discriminator is trained online to maximize the WGAN objective, with updates performed after each individual sample is received during both phases, which are separated in time. Is this an accurate understanding of the proposal? If so, the authors should demonstrate that this approach actually succeeds, as it appears from the methods that their experiments were performed with batch sizes much greater than one. If not, the authors should discuss how the information required to perform batched updates might be stored.

4. In the abstract, introduction, and Figure 1, much is made of the idea that "this manuscript assigns the role of discriminator to particular cell types distributed in the neocortex" (Lines 101-102). It would therefore be helpful to organize the proposed experiments based on which are tests of features particular to the proposed model rather than tests of features shared by related models (or of any model with phase-dependent plasticity). Moreover, the proposed experiments do not appear to address a crucial component of the alternating-phase model: that the connectivity of these interneurons should be determined by the conditional dependence structure of their synaptic partners. Naïvely, this seems challenging to probe. The authors should also elaborate on the question of whether the degree of connectivity required for an effective discriminator neuron is plausible.

5. I do not find the numerical experiments to be well-motivated. Very little justification is given for the widely variable selection of tasks, types of recurrence, and network architectures. As the stated objective of the paper is to address questions relevant to biology, I believe it is important to give at least one proof-of-principle example of an architecture with greater biological motivation. At the very least, the authors should discuss what each experiment reveals, with reference to the paper's overarching goal.

Minor comments

1. The clarity of the figures could be improved significantly. All of the diagrams and illustrations could be made much larger---the labels are currently so small as to be effectively illegible without zooming into the PDF---and additional labels could be added. For example, in Figure 3B, I presume from context that the top rows of the "Reconstruction" and "Denoising" subpanels show input images; this should be explicitly marked.

2. The proposed method for comparing phases of an ascending oscillation seems to resemble at a high level the idea of a stacked GAN, as studied by Huang et al., CVPR 2017 (arXiv:1612.04357). Provided that my understanding is accurate, this and other subsequent works from the machine learning literature should be cited, and the algorithmic (not implementational or interpretational) relationship to the stacked GAN should be highlighted. At the very least, the authors have argued for this setup as a way to tackle networks with recurrence, but I'm not sure how surprised one should be that such a thing would work given the existing literature on GANs.

3. The submitted manuscript is peppered with non-grammatical sentences and distracting typos. To give just two examples:

Lines 101-102: "These proposals share the By contrast, this manuscript assigns the role of discriminator to particular cell types distributed in the neocortex"

Lines 310-311: "In the hippocampus, for example, transitions between potentiation and depression and have been observed with phases of the theta rhythm in hippocampus"

4. In Lines 382-383, the authors write "We emphasize, however, that biology may have found solutions to stabilize this approach" in an effort to excuse the failure of their algorithm to successfully reconstruct MNIST digits. This is neither a necessary comment, nor a satisfactory one. As the network in question features many non-biological features, why appeal to biology here? If they are concerned that this somehow reveals a more fundamental deficiency of their model, this point should be elaborated.

5. The authors mention, but do not discuss in detail, the WGAN algorithm's requirement that the discriminator is Lipschitz. At face value, this constraint would seem to affect the biological plausibility of their proposal, and warrants further discussion.

6. Several papers published in computer science conferences---notably including the WGAN paper, published in ICML 2017---are cited only as arXiv preprints. This should be rectified.

**Have the authors made all data and (if applicable) computational code underlying the findings in their manuscript fully available?**

Reviewer #1: Yes

Reviewer #2: Yes

PLOS authors have the option to publish the peer review history of their article (what does this mean?). If published, this will include your full peer review and any attached files.

Reviewer #1: **Yes: **Jakob Jordan

Reviewer #2: No
---

## [Decision Letter · Decision Letter 1]

24 Jul 2023

Dear Benjamin,

Thank you very much for submitting your manuscript "A role for cortical interneurons as adversarial discriminators" for consideration at PLOS Computational Biology. As with all papers reviewed by the journal, your manuscript was reviewed by members of the editorial board and by several independent reviewers. The reviewers appreciated the attention to an important topic. Based on the reviews, we are likely to accept this manuscript for publication, providing that you modify the manuscript according to the review recommendations.

Two reviewers pointed out several points to improve the clarity of the manuscript, including the section on discussing Wasserstein GAN objective and the Lipschitz constraint. Of importance, is a more detailed comparison with the alternative proposals from Deperrois et al. and Gershman and well as to the method proposed in Belghazi et al.. Commenting on possible experiments that could enable falsification of different theories would be a great strength.

Sincerely,

Pouya Bashivan

Guest Editor

PLOS Computational Biology

Samuel Gershman

%CORR_ED_EDITOR_ROLE%

PLOS Computational Biology

Two reviewers pointed out several points to improve the clarity of the manuscript, including the section on discussing Wasserstein GAN objective and the Lipschitz constraint. Of importance, is a more detailed comparison with the alternative proposals from Deperrois et al. and Gershman and well as to the method proposed in Belghazi et al.. Commenting on possible experiments that could enable falsification of different theories would be a great strength.

Reviewer's Responses to Questions

**Comments to the Authors:**

Reviewer #2: I thank the authors for their effort to address my comments and those of the first referee. I believe the manuscript has substantially improved, and I am now generally in favor of its acceptance as an interesting theoretical proposal.

I have one suggestion, which the authors may consider in preparing the final version of their manuscript: In their discussion of the W-GAN's Lipschitz constraint in Lines 326-333, the authors briefly mention that having bounded synaptic weights is "aligned with natural biological constraints." I think this point is worth expanding upon, as the original W-GAN paper by Arjovsky, Chintala, and Bottou makes the case that "[w]eight clipping is a clearly terrible way to enforce a Lipschitz constraint," which would ideally be replaced with another approach. It would be useful to discuss in greater detail the idea weight clipping is not only biologically reasonable but also potentially (at least to some degree) imposed by biology, ideally with more citations to relevant experimental work.

A few other minor points:

1. In the discussion of the sampling hypothesis (paragraph starting in Line 214), it would be useful to cite Echeveste et al., Nature Neuroscience 2020, and also the original paper by Hoyer and Hyvärinen (NeurIPS 2002).

2. In the first sentence of the abstract, are the words "throughout development" necessary?

Reviewer #3: This is a nice paper exploring the idea that the brain uses adversarial learning techniques to build a statistical model of the world. The paper argues that adversarial learning has a number of appealing aspects that make it a more plausible theory than competing theories based on variational inference. First, recurrent architectures can be handled seamlessly. Second, hierarchical models with local recurrence can be learned with local discriminators. This makes it easier to satisfy the constraints that are known from the architecture of the cortex. The authors make a high-level connection to oscillatory codes to justify a fast alternation between bottom-up and top-down modes, needed to learn the model.

The hypotheses advanced by the paper are interesting and in my opinion sufficiently distinct from prior work by Deperrois et al. and Gershman to warrant publication. There are however certain aspects of the paper that I think could be improved.

My main issues with this paper revolve around clarity. The text mixes and rapidly alternates between discussions of prior machine learning work, biological modeling and plausibility considerations. I understand the nature of the work requires plenty of discussion. But, despite the improvements, there's still too much context switching in the present version. I think that one more iteration of paper reorganization (rather than a rewrite) would be sufficient to improve readability.

I leave below some comments and questions for the authors regarding clarity:

- I found the section "The problem of dimensionality and a solution with oscillations" particularly difficult to follow, and it is a crucial one. I had to read it multiple times to understand whether there was actually a new modeling/algorithmic solution compared to Belghazi et al. 2018. that would then be mapped to oscillations. From what I understand, besides opting for the Wasserstein-GAN/ALI objective, the difference is that here the authors allow for recurrent connections -- within a layer, only. So the essence of the proposed solution is to assume a hierarchical, layered model, and use layer-wise discriminators. The mapping to oscillations comes from the alternation between bottom-up and top-down (fast: say, for every data sample), actually as done by Belghazi et al. too. Is this correct?

- Pseudocode would help.

- A clearer discussion comparing to the proposals of Deperrois et al. and Gershman would strengthen the paper. I'd also like to hear the author's speculation/thoughts on how biology would tackle the fragility issues of adversarial learning.

- Typos should be screened for and corrected.

As for the experiments that the authors present, it seems to me that better understanding how recurrence affects the stability and practical feasibility of adversarial learning is one missing point. (As the authors remark, layerwise recurrence is not a standard feature of current machine learning models.) It might well be that recurrent models allow for more compact, efficient architectures (in loose analogy to e.g. deep equilibrium models, or energy-based models), but are otherwise harder to train. I understand that it's hard to answer such questions, but they would greatly add value to the paper. For instance, it would be interesting, and perhaps not too far-fetched, if recurrence would allow reducing layer size, which should not be too large for the sake of biological plausibility.

**Have the authors made all data and (if applicable) computational code underlying the findings in their manuscript fully available?**

Reviewer #2: Yes

Reviewer #3: None

PLOS authors have the option to publish the peer review history of their article (what does this mean?). If published, this will include your full peer review and any attached files.

Reviewer #2: No

Reviewer #3: No

Figure Files:

Data Requirements:

Reproducibility:

References:

---

## [Editor Report · Decision Letter 2]

31 Aug 2023

Dear Benjamin,

We are pleased to inform you that your manuscript 'A role for cortical interneurons as adversarial discriminators' has been provisionally accepted for publication in PLOS Computational Biology.

Best regards,

Pouya Bashivan

Guest Editor

PLOS Computational Biology

Samuel Gershman

%CORR_ED_EDITOR_ROLE%

PLOS Computational Biology

---

## [Editor Report · Acceptance letter]

13 Sep 2023

PCOMPBIOL-D-22-00990R2 

A role for cortical interneurons as adversarial discriminators

Dear Dr Benjamin,

I am pleased to inform you that your manuscript has been formally accepted for publication in PLOS Computational Biology. Your manuscript is now with our production department and you will be notified of the publication date in due course.

With kind regards,

Anita Estes
